# Coccidia (Apicomplexa: Eucoccidiorida) of Freshwater Fish

**DOI:** 10.3390/microorganisms13020347

**Published:** 2025-02-05

**Authors:** Simuzar Mamedova, Panagiotis Karanis

**Affiliations:** 1Institute of Zoology, Ministry of Science and Education Republic of Azerbaijan, Baku 1073, Azerbaijan; 2Department of Life Sciences, Khazar University, Baku 1001, Azerbaijan; 3Medical Faculty University of Cologne, 50923 Cologne, Germany; karanis.p@unic.ac.cy; 4Department of Basic and Clinical Sciences, University of Nicosia Medical School, Nicosia 2408, Cyprus

**Keywords:** Coccidia, Adeleorina, Eimeriorina, freshwater fish

## Abstract

The phylum Apicomplexa includes endoparasites of fish worldwide, which cause parasitic infections that can adversely affect productivity in aquaculture. They are considered bioindicators of water pollution. Piscine apicomplexan parasites can be divided into two major groups: the intracellular blood parasites (Adeleorina) and the coccidians (Eimeriorina), which can infect the gastrointestinal tract and several organs. This work aims to compile, as completely as possible and for the first time, the available information concerning the species of coccidia (Apicomplexa: Conoidasida), which has been reported from freshwater fish. A comprehensive bibliographic search was performed using all available databases and fields, including Scopus, PubMed, and Google Scholar. In the freshwater fish found, there were 173 described species. This review demonstrates that freshwater fish’s eimeriid coccidia are better studied than adeleid coccidia. Studies of coccidian freshwater fish fauna indicate a high infection with *Eimeria* and *Goussia* species. The wealthiest coccidia fauna were found in the Cypriniformes, Perciformes, Siluriformes and Cichliformes fishes.

## 1. Introduction

The phylum Apicomplexa (Levine et al., 1980) comprises a large group of obligate, intracellular protist parasites that are among the most prevalent and morbidity-causing pathogens worldwide. It includes endoparasites of fish worldwide and is responsible for diseases, which, in turn, may lead to economic and social impacts in different countries. Parasitic infections can adversely affect productivity in aquaculture, and they are considered bioindicators of water pollution. Clinical symptoms of coccidian infection in fish vary from asymptomatic to chronic signs of emaciation, anorexia, listlessness, whitish mucoid feces, abdominal distention, muscle atrophy, retarded growth and increased mortality, depending on the susceptibility of the fish species, fish age and size, cultured conditions, species of coccidia and degree of parasitic infection [1,2].

Freshwater fish are hosts for the many apicomplexan parasites. Apicomplexans have complex life cycles, and there is substantial variation among apicomplexan groups. Both asexual and sexual reproductions may be involved. The basic life cycle may be said to start when an infective stage, or sporozoite, enters a host cell and then divides repeatedly to form numerous merozoites. Apicomplexans are transmitted to new hosts in various ways; some are transmitted by infected invertebrates (leeches or another hematophagous arthropod), while others may be transmitted in the feces of an infected host or when a predator eats infected prey [3,4].

Piscine apicomplexan parasites can be divided into two major groups: the intracellular blood parasites (Adeleorina), which occur in host blood cells and the reticuloendothelial system, and the coccidian (Eimeriorina), which can infect the gastrointestinal tract and several organs.

Adeleorina parasites are currently distributed among six families. Three families of this were reported from fish: Dactylosomatidae (genus: *Dactylosoma* and *Babesiosoma*), Haemogregarinidae (genus: *Haemogregarina*, *Cyrilia* and *Desseria*), and Hepatozoidae (genus: *Hepatozoon*) (see Appendix A, Table A1).

The term “coccidia” has referred mainly to obligate intracellular protozoa of the genera *Eimeria* (Schneider, 1875) and *Isospora* (Schneider, 1881) in the family Eimeriidae, suborder Eimeriorina. According to Pugachev and Krylov, coccidia have been found in the fish, which belong to eight genera. Four genera of coccidia are described and found mainly in fish (*Calyptospora*, *Crystallospora*, *Epieimeria*, *Goussia*). Representatives of the other four genera (*Cryptosporidium*, *Eimeria*, *Isospora*, *Octosporella*) of coccidia parasitize not only fish but also different groups of vertebrates (see Appendix A, Table A1) [5,6,7,8,9].

This work aims to compile, as completely as possible and for the first time, the available information concerning the species of coccidia (Apicomplexa: Conoidasida) that have been reported in freshwater fish.

## 2. Study Design

A comprehensive bibliographic search was performed using all available databases and fields included on Scopus, PubMed, and Google Scholar with the following search string: (Coccidia, Haemococcidia or *Dactylosoma* or *Haemogregarine* or *Babesiosoma*, or *Hepatozoon* or *Cyrilia* or *Calyptospora* or *Cryptosporidium* or *Eimeria* or *Goussia* or *Isospora* or *Epieimeria* or *Crystallospora*) and freshwater fish. Figure 1 illustrates the selection of studies employing the PRISMA flowchart [10]. The study looked for studies using PRISMA standards.

The systematic search used in this study initially turned up 747 documents. Of these articles, 432 were removed based on duplicate records, and 111 of the 315 screened records were disqualified due to their abstract and study similarities. A total of 204 full-length publications were assessed for eligibility, with 53 full-length articles being disqualified for a variety of reasons: old information (28), poor quality (14), and lack of data (11). Finally, 151 publications were included in the review. Here, we review the known best practices of search strategies and provide insight into how well the recent reviews and original research articles studied the top applied coccidians found in freshwater fish, ensuring the reliability and accuracy of our findings.

## 3. Adeleorina

### 3.1. Dactylosoma and Babesiosoma

The genera *Dactylosoma* and *Babesiosoma* belong to the family Dactylosomatidae and include heteroxenous coccidians that cycle between two hosts: an invertebrate and a vertebrate host (Figure 2) [11,12,13].

Members of the family Dactylosomatidae have remained, until recently, some of the most poorly understood members of the phylum Apicomplexa despite the long period that has elapsed since their discovery. Distinctive features between two dactylosomatid genera, *Dactylosoma* and *Babesiosoma*, were identified. However, these genera differ according to the number of merozoits (four merozoits in *Babesiosoma*, more than four in *Dactylosoma*) that formed in the erythrocytes of cold-blooded vertebrates [5,15]. In the definitive host, the number of sporozoites within oocysts of both genera is twice as much as the number of merozoites [13]. The fish host erythrocyte *Babesiosoma* differs from *Dactylosoma* in lacking a nuclear karyosome, possessing vacuolated and only slightly granular cytoplasm, and producing only four merozoites in a rosette or cross [16].

Several related species of this genus *Dactylosoma* were described from cichlids fish (species of *Oreochromis*, *Astatorheochromis* and *Haplochromis*) and grey mullets (Mugilidae—*Mugil cephalus*, *Liza richardsoni*, *L. dumerili*) in Africa throughout the first half of the XX century. The first species of *Dactylosoma* to be reported from fish was described by Hoare (1930) in *Haplochromis* spp. of Lake Victoria, Uganda, and named *Dactylosoma mariae* [17]. Jakowska & Nigrelli (1956) separated *D. mariae* Hoare, 1930 and placed them under the new genus *Babesiosoma* [15]. *D. mariae = B. mariae* Hoare, 1930; Baker, 1960 were found in Lakes Victoria and George, with a single record from *O. mossambicus* in Transvaal, South Africa. This parasite reported from the Okavango Delta (Botswana) produces four intraerythrocytic merozoites from cruciform meronts in the fish species *Serranochromis angusticeps* [18].

*Dactylosoma hannesi* (Paperna, 1981) was discovered in the blood erythrocytes of *Mugil cephalus* Flathead grey mullet), *Chelon richardsonii* (South African mullet) and *C. dumerili* (Grooved mullet) from Swartkops estuary, located east of Port Elizabeth, South Africa [19]. Recent studies indicate that *D. hannesi* is a *Babesiosoma* species, concluding that *B. mariae* and *B. hannesi* are synonymous [13,20].

The other species of the genus *Dactylosoma* to be recorded from fish is *D. salvelini* Fantham, Porter & Richardson (1942) from the Canadian trout (*Salvelinus fontinalis*) from the Province of Quebec, Canada. This species produced eight merozoites slightly smaller than *D. mariae* [21]. Sounders (1960) described *D. lethrinorum* from a Spangled emperor (*Lethrinus nebulosus*) and Pink ear emperor (*L. lentjan*) from the water of the Red Sea near Al Gardaga, Egypt [22].

All of these studies belong to the XX century. In recent years, two works have documented the presence of *Dactylosoma* in African regions [23,24].

Two additional dactylosomatid species from Indian fishes have been incorrectly assigned to the genus *Dactylosoma*. *D. striata* Sarkar and Haldar, 1979 and *D. notopterae* Kundu and Haldar, 1984 were described from the erythrocytes of freshwater teleosts [13,25]. Both species more closely resemble *Babesiosoma* species than *Dactylosoma* species, leading to the conclusion that *D. striata*, *D. notopterae* and *B. ophicephali* are synonymous [13].

As we noted, compared to species of *Dactylosoma*, species of *Babesiosoma* produce only four merozoites during each merogonic cycle and eight sporozoites within oocysts in the definitive host [13]. The *Babesiosoma* species described have a cosmopolitan distribution in marine and freshwater environments [13].

The first described *Babesiosoma* species was *Babesiosoma mariae* Hoare, 1930 (=*B. hannesi* Paperna, 1981). *B. mariae* were also detected from freshwater fish *Chrysichthys auratus* at Al Fath Center in Assiut Governorate, Egypt [26]. *B. bettencourti* França, 1908 (syn. *Haemogregarina bettencourti* França, 1908; *Desseria bettencourti* Siddall, 1995) were recorded from the erythrocytes the European eels, *Anguilla anguilla* L., captured in six rivers in northern and central Portugal [27]. *B. tetragonis* was observed in the erythrocytes of *Catostomus* sp. from the Shasta River north of California [28]. *B. ophicephali* and *B. hereni* are described from the red blood cells of the freshwater teleost fish *Ophicephalus punctatus* (=*Channa punctatus* (Spotted snakehead)), collected from the suburbs of Calcutta, West Bengal, India [29,30]. *B. batrachi* is described from the erythrocytes of a freshwater teleost fish, *Clarias batrachus* (Philippine catfish), from West Bengal, India [31]. *B. rubrimarensis* Saunders, 1960 was recorded from six specimens of fish (*Lethrinus xanthochilu* (Yellowlip emperor), *L. variegatus* (Slender emperor), *Cephalopholis miniatus* (Coral hind), *C. hemistictus* (Yellowfin hind), *Scarus harid* (Candelamoa parrotfish), *Mugil troscheli =* (*Mugil poecilus* (Largescale mullet)) from the water of the Red Sea near Al Gardaga, Egypt (see Appendix A, Table A2) [22].

In recent years, *Babesiosoma* sp. was recorded from two Egyptian freshwater fish species from the orders Cypriniformes and Siluriformes: *Cyprius carpio* (Common carp) and *Clarias gariepinus* (African catfish) [32] and three species from the order Cypriniformes of freshwater fish (*Barbus grypus* (Shabout), *B. sharpeyi* (Bunnei), *Carasobarbus luteus* (Yellow barbell) collected from the Khazar River in Ninevah Governorate, Iraq [33]. *Babesiosoma* sp. was also found in Cypriniformes fish—*Triplophysa marmorata* (Kashmir triplophysa-loach) from Anchar Lake and River Jhelum in Kashmir [34].

In conclusion, currently, there are five recognized species of *Dactylosoma*, two of which infect fish hosts, namely *D. lethrinorum* Saunders, 1960 and *D. salvelini* Fantham, Porter and Richardson, 1942 [24]. There are currently seven recognized species of *Babesiosoma* described from fish hosts, each with unique characteristics and adaptations. Five of them, namely *B. mariae* Hoare, 1930 or *B. hannesi* Paperna, 1981; *B. tetragonis* Becker and Katz, 1965; *B. ophicephali* Misra et al., 1969; *B. hareni* Haldar et al., 1971; and *B. batrachi* Haldar et al., 1971, have been recorded from freshwater fish (see Appendix A, Table A2).

### 3.2. Haemogregarina, Cyrilia, and Desseria

Haemogregarines are broadly distributed among vertebrate hosts, including fishes [6]. The development of haemogregarines is complex and occurs through two types of hosts: blood-sucking invertebrates (leeches) and vertebrates, including fish [4]. In haemogregarines, merogony and transformation of merozoites into gamonts occur in the blood cells of fish. In contrast, the differentiation of gamonts into gametes, zygote production and sporulation resulting in oocyst formation occur in leech vectors [5]. Oocysts of the genus *Haemogregarina* contain eight or more naked sporozoites [6].

Siddall (1995) placed fish haemogregarines in the genera *Desseria* Siddall, 1995 (41 species), *Cyrilia* Lainson, 1981 (1 species), and *Haemogregarina* Danilewsky, 1885 (sensu lato) (13 species) [35,36,37,38,39,40,41,42,43,44,45,46,47,48,49]. They are commonly found in both erythrocytes and leukocytes of marine fishes, except for *Cyrilia* sp., which parasitize only the erythrocytes of freshwater fish [4]. Members of the three genera are differentiated by their development in vertebrate and invertebrate hosts. *Cyrilia* and *Haemogregarina* are both characterized by an intra-erythrocytic merogony phase in fish hosts, while in *Desseria* sp., this stage is not found [50]. The life cycles of a few fish haemogregarines are known, including *Cyrilia*, *Desseria* and some *Haemogregarina* sp.; they employ leeches as definitive hosts [49,51].

Haemogregarines were first recorded from the blood of marine fishes in France by Laveran and Mesnil (1901). They recorded *Haemogregarina simondi* in Dover sole, *Solea solea* (=*Solea vulgaris)*, caught in the English Channel, and *H. bigemina* in blennies *Lipophrys pholis* (=*Blennius pholis)* and *Corypho-blennius galerita* (=*Blennius montagui*) (see Laveran and Mesnil, 1902) at Cap de la Hague in France (Davies, 1995). Most haemogregarines have been recorded from marine fish [4,52]. One of the most prevalent and cosmopolitan species of haemogregarines is *H. bigemina*, a marine fish parasite. *H. bigemina* Laveran et Mesnil, 1901 was recorded from 96 species of marine fishes across 70 genera and 34 families [50].

Wenyon (1908) was the first to describe a haemogregarine from a freshwater fish, *H. nili* from *Ophiocephalus obscurus*, in the Nile River, Egypt [35]. In 1923, Franchini and Saini described from perch (*Perca fluviatilis*) in France the species *H. percae*, along with three other new species from common freshwater fishes. It is estimated that some apicomplexans, out of 23 *Haemogregarina* species that are described in freshwater fish, are later placed in *Cyrilia*, *Hepatozoon* or other genera [4].

Other species that infect freshwater fish include *H. catostomi*, *H. vltanensis*, *H. majeedin*, and *H. daviesensis*: *H. catostomi* Becker 1962 occurred in largescale sucker (*Catostomus macrochelius*) and bridge lip sucker (*C. columbianus*), fishes of the central Columbia River [35,36]. *H. vltavensis* Lom et al., 1989 was observed from the red blood cells of the blood of perch (*Perca fluviatilis*) in southwestern Czechoslovakia [37]. *H. majeedin* Al-Salim, 1993 was reported and described from the heart blood film of the freshwater fish Bunnei (*Barbus sharpeyi*) captured in the Shalt al-Arab River, Basrah, Iraq [38]. *H. daviesensis* Esteves-Silva et al., 2019 is described from South American lungfish (*Lepidosiren paradoxa*) in the eastern Amazon region (see Appendix A, Table A2) [39].

*H. cyprini* Smirnoviç 1971 was reported from the blood of *Cyprinus carpio* (order Cypriniformes) in Iraq [40], *H. meridianus* Al-Salim, 1989 was described from the erythrocytes, erythroblasts and blood plasma of *Planiliza abu* (reported as *L. abu*) (order Mugiliformes) from the Shatt Al-Arab river [41]. *H. acipenseris* was recorded in the Siberian starlet (*Acipenser ruthenus marsiglii*) (order: Acipenseriformes) in the rivers of the Lower Irtysh basin [42]. This species lacked morphometric parameters.

*Haemogregarina* sp. was detected in the blood, liver, ovaries and kidneys of *Glarias gariepinus* (order: Siluriformes) in Sudan [43], in blood samples and the blood within the gill tissue of *Oreochromis niloticus* (order: Cichliformes) in Egypt [44] and in the erythrocytes of four specimens of freshwater Cypriniformes fish (*Barbus grypus* (Shabout), *B. sharpeyi* (Bunnei), *Carasobarbus luteus* (Yellow barbell)) and Mugiliformes fish (*Liza abu* (=*Planiliza abu*) (Abu mullet)) collected from the Khazar River in Ninevah Governorate, Iraq [33]. Intra-erythrocytic stages of *Haemogregarina* sp. were observed in *Neoarius graffiti* (Blue salmon catfish) (order: Siluriformes) sampled from the Brisbane River, Western Australia [45]. Haemogregarines are also found in the erythrocytes of the Kessler’s sculpin *Leocottus kesslerii* in Lake Baikal [46].

The genus *Cyrilia* was created in 1981. Species of *Cyrilia* infect freshwater fishes and are transmitted by the bite of African fish leeches, *Batracobdelloides tricarinata*. Hosts include the African catfish (*Clarias gariepinus*) and Nile tilapia (*Tilapia nilotica*). The genus *Cyrilia* is of ecological interest, as it affects species such as *Potamotrygon wallacei* (Cururu stingray), found in the Amazon region, that help control the invertebrate population and act as bioindicators in environmental monitoring studies [47].

Lainson (1981) described *Cyrilia gomesi* parasitizing the freshwater swamp eel (*Syngamus marmoratus*) from Para State, North Brazil [53]. *C. gomesi* (Lainson 1981) was later amended to *C. lignieresi* (Lainson 1992) [6,53,54]. In previous work, Diniz et al. analyzed the ultrastructure of *C. lignieresi* trophozoites by transmission electron microscopy [7]. Some morphological characteristics of *C. lignieresi* have been described previously, but the parasite–erythrocyte relationship is still poorly understood [55]. *Cyrilia* sp. was recorded, and the trophozoites and all stages of gamonts were well described in the freshwater fish Cururu stingray from Rio Negro, a river of Amazonas, Brazil (see Appendix A, Table A2) [47].

The genus *Desseria* was described in 1995. All currently recognized species in this genus are infected fish. Species of *Desseria* were described very poorly; only vertebrate stages are known, and often, they are not described in detail. Some formerly known *Deseria* species are related to other genera [6,56,57]. Contemporary literature has insufficient information about *Deseria*. In 2021, Quraishy et al. suggested the family Haemogregarinidae, which contains the genera *Haemogregarina* and *Cyrilia* [58].

Many species of haemogregarines of fish are described in the old literature [4,21]. These works are problematic because of the tendency for brief or incomplete descriptions. For most of the first half of the XX century, there was a tendency to describe species of haemogregarines exclusively based on the intraerythrocytic gamont. Indeed, authors named new species for nearly every haemogregarine discovered in a novel host species or geographical location. By the middle of the XX century, several authors agreed that describing species based on these features or the structure of the bloodstream gamonts alone was specious [59]. For this reason, we prefer the new information about haemogregarines. Only one study using molecular tools to determine the species of haemogregarines in freshwater fish has been reported [58].

### 3.3. Hepatozoon

*Hepatozoon* is a genus of apicomplexan parasites known to cause musculoskeletal disease in numerous terrestrial vertebrate species whose life cycle involves sexual development (sporogony) in definitive hosts that are hematophagous arthropods (ticks, mites, lice and tsetse flies) and asexual development (merogony in visceral tissues; gamontogony in leukocytes) in a vertebrate host [59].

Species of *Hepatozoon* have been described from all groups of vertebrates, but their occurrence in fish is an infrequent phenomenon. Only two studies report the occurrence of parasites belonging to the *Hepatozoon* genus in fish [48,60]. Cysts containing *Hepatozoon* cystozoites were observed in the liver of *Hoplias aimar* (Poisson-tigre géant) (order: Characiformes) from the Eastern Amazon. This unique finding, the first report of the occurrence of cysts containing *Hepatozoon* cystozoites in free-living fishes, was confirmed by molecular analysis. The sequencing revealed that *Hepatozoon* sp. has identity with *Hepatozoon caimani*, detected in caimans (*Caiman yacare*) from Brazil [48]. Previously, it was experimentally demonstrated that *Metynnis* sp. fish are susceptible to *H. caimans* and develop numerous cysts morphologically similar to those described by Lainson et al. (2003). *Metynnis* sp. successfully transmitted *H. caimans* to caimans [60].

## 4. Eimeriorina

In a systematic respect, fish coccidia (Eimeriorina) have been studied much better than the intracellular blood parasites (Adeleorina). The life cycles of the coccidia that infect fish can be divided into two principal types—monogenic (the cycle occurs in one host; *Cryptosporidium*, *Eimeria*, *Goussia*, *Isospora*, *Epieimeria*, *Crystallospora*, and *Octosporella*) and heterogenic (involving both intermediate and paratenic hosts; *Calyptospora*) (Figure 3 and Figure 4).

They parasitize different organs in vertebrate hosts. Oocysts of coccidian parasites were found in the epithelium of the gut, stomach, liver, kidney, spleen, gonads, gallbladder and feces [3]. The structure of oocysts and sporozoites is important for the taxonomy of these parasites (see Appendix A, Table A1) [5].

Coccidia in fish was first discovered and described 110 years ago. Several reviews on the regional and world fauna of coccidia of fishes were published during that period. Pellerdy (1974) and Dykova and Lom (1983) published the most significant reports on the world fauna of coccidia. Pellerdy (1974), in the monograph “Coccidia and Coccidiosis”, gives the names of 66 species of coccidia in infected fishes. Dykova and Lom (1983) described as many as 127 species of coccidia in fishes [3,61].

### 4.1. Calyptospora

The genus *Calyptospora* was erected in 1984 by Overstreet et al. to encompass species with sporocysts, without Stieda or sub-Stieda bodies, with a veil supported by sporopodia, and with an anterior apical opening [62]. The genus *Calyptospora* has heteroxenous life cycles involving fishes and shrimp. These intracellular protozoan parasites are found in the liver and intestine of their fish hosts. They have a heteroxenic life cycle transmitted by an infected crustacean (shrimp) ingested by a fish. In fish hosts, the oocysts (elliptical, oval or pear-shaped) present four sporocysts covered by a thin veil fixed by the presence of wall projections named sporopodia. Moreover, it presents a suture on the walls that does not divide the cell into two valves [63].

The first described species of *Calyptospora* was *Calyptospora funduli*, which lived in marine fishes [64]. The species diversity of *Calyptospora* is still poorly known; only six species included in the genus *Calyptospora* are described, especially fish parasitic parasites in the tropical Amazon region: *C. serrasalmi* Cheung, Nigrelli and Ruggieri, 1986 from piranha *Serrasalmus niger* and *S. rhombeus* [65,66]; *C. tucunarensis* Békési and Molnár, 1991 from tucunaré *Cichla ocellaris* [67]; *C. spinosa* Azevedo, Matos and Matos, 1993 from joaninha *Crenicichla lepidota* [68]; *C. paranaidji* Silva et al., 2019 from cichlid fish *Cichla piquiti* [69]; and *C. gonzaguensis* Silva et al., 2020 from Dusky Narrow Hatchetfish *Triportheus angulatus* [70]. Only one species—*C. empristica* Fournie et al., 1985—was described from the freshwater starhead topminnow, *Fundulus notti*, in southern Mississippi (North America) (see Appendix A, Table A3) [71].

There are also records of the parasitism of *Calyptospora* sp. in Brazilian freshwater fish—*Arapaima gigas* (Arapaima, Osteoglossiformes) [72], *Triportheus guentheri* and *Tetragonopterus chalceus* (both of Characiformes) [73], *Brachyplatystoma vaillantii* (Laulao catfish, Siluriformes) [74], *Cichla temensis* (Speckled pavon, Cichliformes) [75], and *Aequidens plagiozonatus* (Cichliformes) (see Appendix A, Table A3) [76].

### 4.2. Cryptosporidium

The apicomplexan parasite *Cryptosporidium* is a significant intestinal pathogen causing acute diarrheal disease in various vertebrates, including humans [77,78,79,80,81]. Its infections are not confined to a specific region, as they occur globally, and are emerging as a disease in both marine and freshwater fish in numerous countries. *Cryptosporidium* species have been reported in 23 freshwater and 24 marine fish species [82], highlighting the global prevalence of this zoonotic risk. Detecting the protozoa in fish is a reliable indicator of environmental contamination or ecosystem health, suggesting a potential environmental source of human exposure and infection. The parasite is transmitted through various routes, including ingesting contaminated food or water, person-to-person contact, contact with animals, and recreational water, all contributing to the fecal–oral contamination route [83].

*Cryptosporidium* has some features that differentiate them from all other Coccidia, including (1) intracellular and extra-cytoplasmic localization, (2) forming of a “feeder” organ, (3) presence of morphological (thin- or thick-walled) oocysts as well as functional (auto vs. new infection) types of oocysts, (4) small size of oocysts, (5) missing some morphological characteristics, such as sporocysts or micropyles, and (6) the resistance of *Cryptosporidium* to all the available anti-coccidial drugs [84].

The traditional classification of *Cryptosporidium* within the coccidians has now been securely rejected based on comparative ultrastructural and genomic data. Based on recent genetic analyses, *Cryptosporidium* is no longer considered a coccidian and has been reclassified as a gregarine within the subclass Cryptogregaria. Like the gregarines, *Cryptosporidium* exhibits plasticity in its life cycle and options for parasitism, including the ability to multiply without host cell encapsulation. Different reports have supported the similarities between *Cryptosporidium* and gregarines. However, the position of the genus *Cryptosporidium* on the evolutionary tree of Apicomplexa is unresolved [85,86,87].

In contrast with the epicellular location of *Cryptosporidium* species from other vertebrates, in the case of piscine *Cryptosporidium* species, sporulation occurs deep within the epithelium in the piscine *Cryptosporidium* species [88,89,90].

The environmental stage, the oocyst, is environmentally robust, and there are few distinguishing morphological features; therefore, molecular characterization is required to delimit species. Currently, 44 species of *Cryptosporidium* are recognized based on molecular and biological criteria [91]. Five species of them have been genetically described in fish as a specific host: *C. molnari* Álvarez-Pellitero & Sitjà-Bobadilla 2002, *C. scophthalmi* Álvarez-Pellitero et al., 2004, *C. huwii* Ryan et al., 2015, *C. bollandi* Bolland et al., 2020 and *C. abrahamseni* Zahedi et al., 2021 [82].

The first piscine host of *Cryptosporidium* found was the tropical marine fish *Naso lituratus*, in which *C. nasorum* was identified by Hoover in 1981 (see [92]). After the first description, different histological studies reported developmental stages of *Cryptosporidium* in several marine and freshwater fish species’ stomachs and intestines [2]. The first report on *Cryptosporidium* of freshwater fish was published in 1983 [92]. Currently, the following three species are recognized as specific to freshwater fish hosts: *C. huwii* from guppy (*Poecilia reticulata*), golden tiger barb (*Puntigrus tetrazona*) and neon tetra (*Paracheirodon innesi*) [89,93,94]; *C. bollandi* from angelfish (*Pterophyllum scalare*) and Oscar fish (*Astronotus ocellatus*) (Bolland et al., 2020); and *C. abrahamseni* from red-eye tetras (*Moenkhausia sanctaefilomenae*) [95].

*C. molnari* and *C. scophthalmi* are described from marine fish [88,96,97]. Some freshwater fish species have been reported as other natural hosts for *C. molnari* (see Appendix A, Table A4) [98,99,100,101,102].

*Cryptosporidium* species found in other groups of vertebrates have also been identified in freshwater fish: *C. parvum*, *C. hominis* and rat genotype III [100,101,103,104,105,106,107,108,109,110,111,112]. Current knowledge of the epidemiology, taxonomy, pathology and host specificity of *Cryptosporidium* species infecting piscine hosts is higher than in previous years. Furthermore, many different genotypes and *Cryptosporidium* sp. have been identified in fish (see Appendix A, Table A4).

### 4.3. Eimeria, Goussia, Isospora, Octosporella

The Coccidia genus, which contains many species that infect a wide variety of fish, reptiles, birds and mammals, is a fascinating area of study due to its host-specific nature [113,114,115,116,117,118,119,120,121,122]. Almost all species are specific to a particular host. The life cycle of *Eimeria* and all other species is considered monoxenous, meaning that the cycle occurs in one host. The life cycle of the specimens of this genus has two distinct stages: the exogenous phase (sporogony) and the endogenous phase (schizogony and gametogony). Fish *Eimeria* species differ from typical *Eimeria* species from higher vertebrates in having, as a rule, a thin oocyst wall and endogenous sporulation [123].

In the *Eimeria* genus, four sporocysts develop within the circumplasm of the oocyst, each containing two banana-shaped sporozoites. The genus *Goussia* was erected to accommodate piscine coccidians with oocysts possessing four sporocysts, with two valves joined by a longitudinal suture. *Eimeria* and *Goussia* of freshwater fishes from various parts of the world have been studied relatively well (see Appendix A, Table A5 and Table A6). Table A5 lists 88 species of the *Eimeria*, and Table A6 lists 52 of the *Goussia* reported from freshwater fish.

Significant findings have been made in recent years in studying coccidians in various fish species. *Eimeria* sp. was recorded from the freshwater fish *Liza abu* = *Planiliza abu* (Abu mullet) obtained from Tigers River in Mosul city, Iraq [118]. In the last years, *Eimeria* sp. and *Isospora* sp. were recorded in Nile tilapia (*Oreochromis niloticus*) in Upper Egypt [124].

*Eimeria* species were detected in Siluriformes fish—*Clarias gariepinus* and Heteroclarias species from some selected fish farms in Kaduna state, Nigeria [125,126,127,128,129,130,131,132,133,134,135]—*Astronotus ocellatus* (Oscar, Cichliformes) and Cypriniformes fish—*Carassius auratus* (Goldfish) *Puntius conchonius* (Rosy barb) in Iran [120].

Most of the species reported in freshwater fish belong to *Eimeria* and *Goussia*, and only a handful belong to the genera *Isospora*, *Epieimeria*, *Crystallospora* and *Octosporella*. Only two freshwater fish species have been found—*Isosopa lotae* Belova, Krylov, 2001 and *I. sinensis* Chen, 1984. Three species of the genus *Octosporella* have been described from Cyprinid fish from Lake Sasajewun and Lake Opeongo, Algonquin Park, Ontario, North America [136,137,138,139,140,141,142,143,144,145] (see Appendix A, Table A5, Table A6 and Table A7). More information about *Octosporella* species in fish needs to be provided in the latest literature. *Epieimeria* and *Crystallospora* species have been recorded in marine fish [146].

## 5. Distribution of Coccidian Freshwater Fish

### 5.1. Distribution of Coccidia by Order of Freshwater Fish

Fish species worldwide are classified into 93 orders [147]. Our search identified 200 defined and undefined species of adeleid and eimeriid coccidians distributed among 11 genera and found in representatives of only 22 orders of freshwater fish: *Dactylosoma*, *Babesiosoma*, *Haemogregarina*, *Cyrilia*, *Hepatozoon*, *Calyptospora*, *Cryptosporidium*, *Eimeria*, *Goussia*, *Isospora* and *Octosporella* (see Appendix A, Table A8).

In freshwater fish, 173 described species have been identified: *Cyrilia* and *Hepatozoon*, each with one described species; *Dactylosoma* and *Isospora*, both with two described species; *Octosporella*, with three described species; *Babesiosoma*, with five described species; *Cryptosporidium*, with six described species (four specific to piscine hosts and two specific to mammals); *Calyptospora*, with six described species; *Haemogregarina*, with seven described species; *Goussia*, with 52 described species; and *Eimeria*, with 88 described species (see Appendix A, Table A2, Table A3, Table A4, Table A5, Table A6 and Table A7).

For some species, obtaining information regarding the characteristics considered in this review was not possible. Most of the descriptions of *Eimeria*, *Goussia*, *Calyptospora*, *Octosporella* and *Isospora* species have relied only on the morphology of oocysts and their location in the hosts. Although fish coccidians began to be described more than one century ago, genetic data on fish coccidians have only started to be collected since the 2010s [105,141,148]. Genetic data are mainly missing for piscine Coccidia (freshwater and marine), except for *Cryptosporidium* species [149,150].

In the representatives of the order Cypriniformes, Perciformes, Cichliformes, Siluriformes, Salmoniformes, Characiformes, Cyprinodontiformes, Asipenceriformes, Anabantiformes, Eupercaria and Mugiliformes, both adeleid and eimeriid coccidia have been found. In the orders Ceradontiformes, Synbranchiformes and Myliobatiformes, adeleid coccida have been found; and in fishes of the order Cyprindontiformes, Esociformes, Centarchiformes, Gobiiformes, Gymnotiformes, Osmeriformes, Beloniformes, Gadiformes and Osteoglossiformes, only eimeriid coccidia have been described. Representatives of these orders are characterized by low abundance and few species (see Appendix A, Table A8). From our point of view, this circumstance explains their absence in the description of coccidia.

The wealthiest coccidia fauna were found in the Cypriniformes, Cichliformes Siluriformes and Perciformes fish. Thus, coccidia belonging to seven genera (*Babesiosoma*, *Haemogregarina*, *Cryptosporidium*, *Eimeria*, *Goussia*, *Isospora* and *Octosporella*) were found in fishes of the order Cypriniformes. Six genera of coccidia (*Babesiosoma*, *Haemogregarina*, *Calyptospora*, *Cryptosporidium*, *Eimeria* and *Isospora*) were described in fishes of the order Cichliformes, and the same number of genera were represented in fishes of the order Siluriformes (*Babesiosoma*, *Haemogregarina*, *Calyptospora*, *Cryptosporidium*, *Eimeria* and *Goussia*). Coccidia belonging to three genera have been described as representatives of three orders: Perciformes, Charasiformes and Mugiliformes. In the five orders (Salmoniformes, Cyrinidontiformes, Centarchiformes, Anabantiformes, Asipenceriformes), coccidia were found belonging to three genera. In the remaining 11 orders (Esociformes, Gobiiformes, Eupercaria, Gymnotiformes, Osmeriformes, Gadiformes, Ceradontiformes, Osteoglossiformes, Beloniformes, Synbranchiformes, Myliobatiformes), coccidia were found belonging to either one or two genera (see Appendix A, Table A8).

The results of the analysis of various taxonomic groups of coccidia in fishes have appeared rather attractive. Coccidia of the genera *Cryptosporidium*, *Eimeria* and *Goussia* were found in fishes most frequently (Figure 5). *Cryptosporidium* species were found in different groups of coccidians, especially in the orders Cypriniformes and Chichliformes. Most *Eimeria* and *Goussia* species were found in the orders Cypriniformes and Perciformes [151,152]. The genera *Dactylosoma*, *Cyrilia*, *Hepatozoon*, *Isospora* and *Octosporella* have limited fish distribution. Coccidia of the genera *Hepatozoon* occurred only in Characiformes and *Octosporella,* only in Cypriniformes (see Appendix A, Table A9 and Table A10).

Common carp (*Cyprinus carpio*) and Goldfish (*Carassius auratus*) in the order Cypriniformes, European perch (*Perca fluviatilis*) and Bullhead (*Cottus gobio*) in the order Perciformes, North African catfish (*Clarias gariepinus*) in the order Siluriformes and Peacock cichlid (*Cichla ocellaris*) in the order Chichliformes were the most infected fish species (Figure 6). The distribution of systematic groups of coccidia by fish order and fish species is apparently determined by two main factors: the first is the quantitative and qualitative composition of fish orders, and the second is the intensity of the survey.

### 5.2. Distribution of Coccidian Freshwater Fish by Continent

Current knowledge on the geographic distribution of freshwater fish adeleid coccidia is limited. Most species of the genus *Babesisoma* have been reported from Asia and Africa. *Haemogregarine* infections of freshwater fish occurred in all continents. Asia is the primary source of information concerning the haemogregarine infection of freshwater fish. Little information has been published about *Dactylosoma*, *Cyrilia* and *Hepatozoon* infection of freshwater fish in Africa and South America (see Appendix A, Table A8).

Eimeriid coccidia has a more prevalent geographical distribution in freshwater fish than adeleid coccidia. The species of *Eimeria* and *Goussia* are widespread all over the world. All continents except South America have had descriptions of them. Most species of *Eimeria* and *Goussia* have been reported from Eurasia and North America; a significant fraction have also been reported from South America, Africa and Australia (see Appendix A, Table A9 and Table A10).

*Cryptosporidium* infections of freshwater fish have been documented from all continents. Almost all described species of *Cryptosporidium*, and the vast majority of them, have been reported from Australia. A significant percentage of *Cryptosporidium* infections in freshwater have been reported in Eurasia. There is a lack of knowledge about *Cryptosporidium* infections in freshwater fish in America and Africa (see Appendix A, Table A9 and Table A10).

All species of *Calyptospora* and *Octosporella* of freshwater fish have been reported from South America (Amazon) and North America, respectively. Little information is available about *Isospora* infection of freshwater fish in Eurasia and Africa (see Appendix A, Table A9 and Table A10).

## 6. Conclusions

This review demonstrated that freshwater fish’s eimeriid coccidia are better studied than the adeleid coccidia of freshwater fish. Studies of the coccidian fauna of freshwater fish indicate that infection is highly prevalent among the *Eimeria* and *Goussia* species. The wealthiest coccidia fauna were found in the families of valuable commercial fish such as Cypriniformes, Perciformes, Siluriformes and Cichliformes fish. Fish coccidiosis is an essential and most prevalent protozoan parasitic disease that affects fish economic loss in aquaculture due to the morbidity and mortality rates reported in some fish species. For the control and prevention of fish coccidiosis, proper hygienic and bio-security measures should be implemented to remove the coccidia oocysts from aquaculture facilities, where fish cohabit in dense groups and are subjected to other stress factors that can enhance the transmission of these parasites.

## Figures and Tables

**Figure 1 microorganisms-13-00347-f001:**
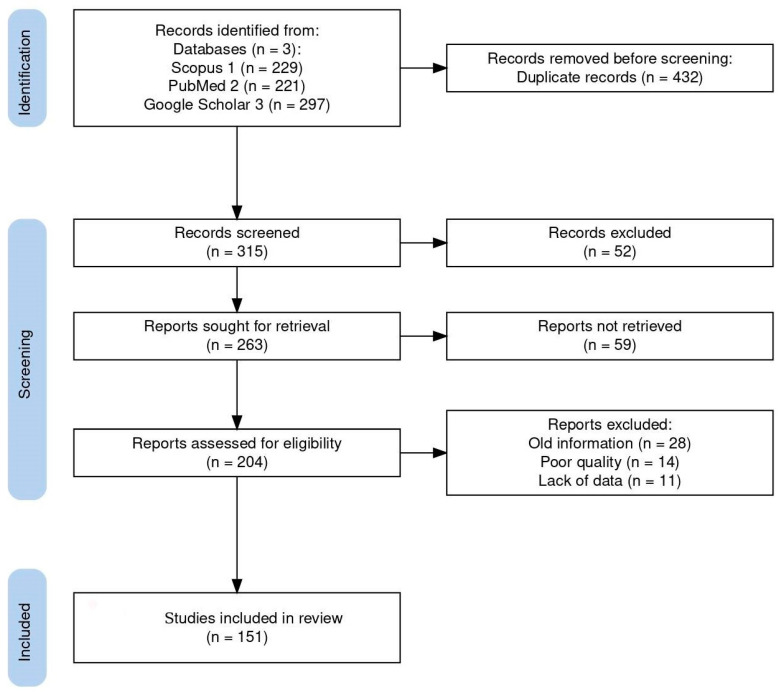
Flow diagram of the study design process showing the total number of retrieved reports, number of records after duplicates were removed, number of records screened and excluded, number of full-text articles assessed for eligibility, number of selected articles for analysis, and number of studies included in the review.

**Figure 2 microorganisms-13-00347-f002:**
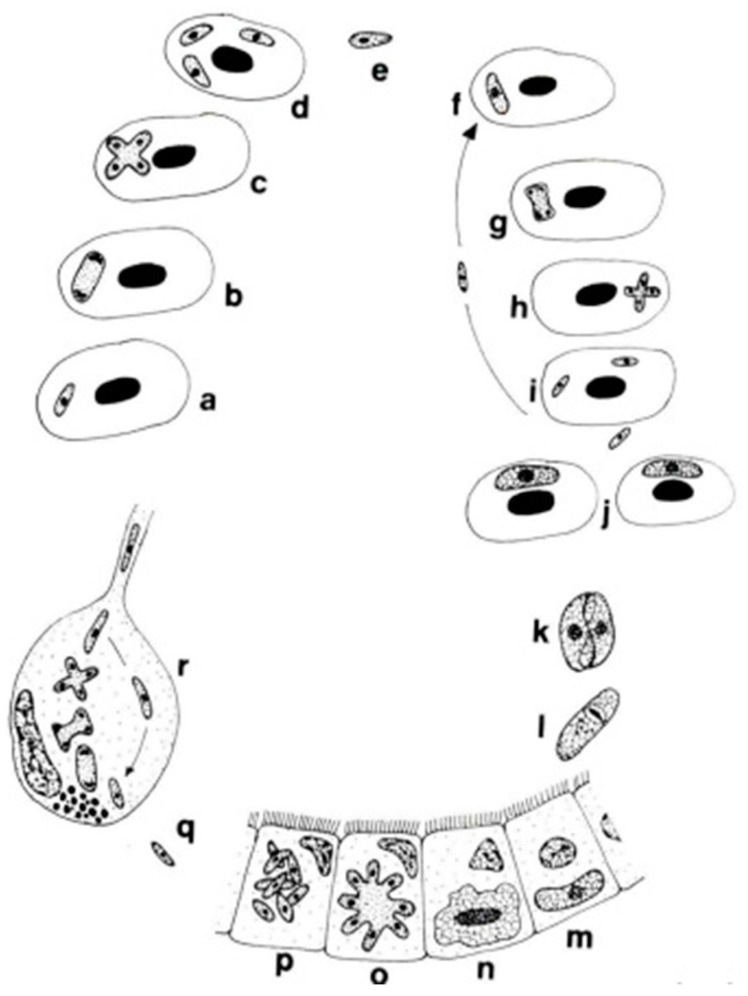
Life cycle of *Babesiosoma stableri*: a–e. Primary intraerythrocytic merogony; f–i. secondary merogony; j. gamonts; k–l. gamonts, freed in the blood meal, associate in syzygy, mature, and fuse to form a zygote; m. zygote (ookinete) penetrates the intestinal epithelial cells; n–p. sporogony forming eight naked sporozoites; q. migration of sporozoites to salivary cells; r. merogony in salivary cells produces numerous infective merozoites injected into the next host when the leech feeds (based on Barta and Desser, 1989) [14].

**Figure 3 microorganisms-13-00347-f003:**
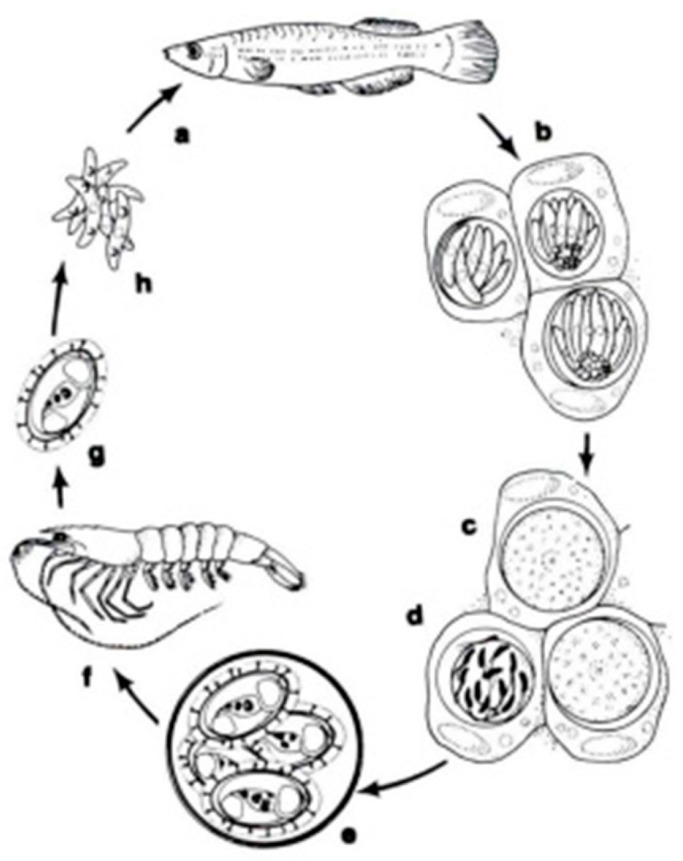
The life cycle of *Calyptospora funduli*: a. sporozoites ingested along with shrimp by topminnows; b. sporozoites enter hepatocytes and undergo two generations of merogony: c. macrogametes and d. microgametocyte with microgametes; e. oocysts develop from zygotes and sporulate in situ; f. oocysts in the liver of decaying fish ingested by shrimp; g. liberation of sporocysts in the gut of shrimp; h. sporozoites enter the hepatopancreas and undergo a period of maturation [14].

**Figure 4 microorganisms-13-00347-f004:**
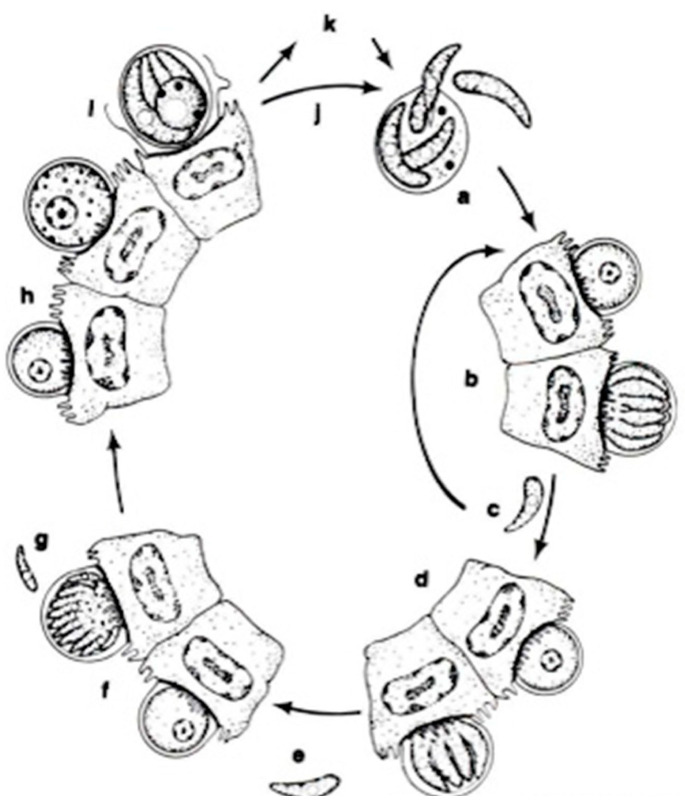
The life cycle of *Cryptosporidium parvum*: a. oocyst containing four sporozoites excysting in intestinal tract; b. type I meront; c. type I merozoites either recycle to produce additional type I meronts (b) or produce type II meronts; d. type II meront; e. type II merozoites penetrate new cells to form male and female gametes; f. microgametocyte; g. microgamete; h. macrogamete; i. sporulated oocyst in situ; j. thin-walled oocysts exocyst in situ (a) and infect new cells; k. thick-walled oocysts pass out in feces into the environment [14].

**Figure 5 microorganisms-13-00347-f005:**
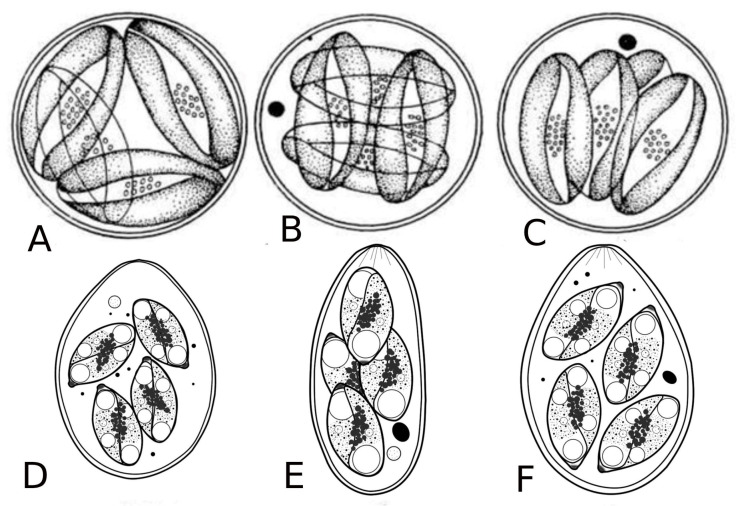
Oocysts of coccidia of freshwater fish. A, B, C—oocysts of *Goussia*; D, E, F—oocysts of *Eimeria* [9].

**Figure 6 microorganisms-13-00347-f006:**
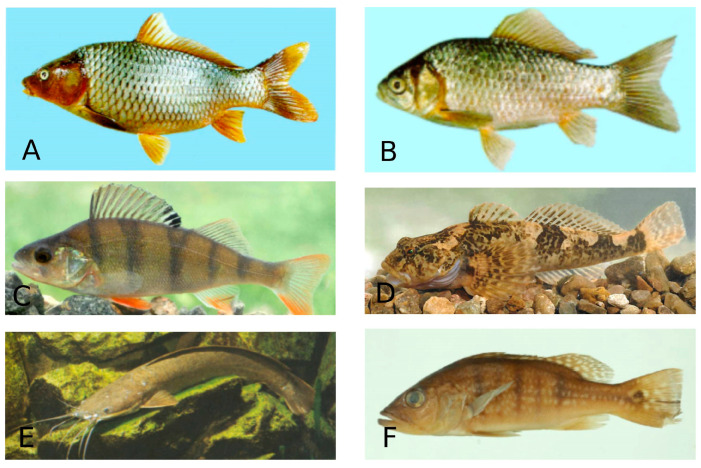
Species of freshwater fish. A—*Cyprinus carpio* (Common carp)—Cypriniformes; B—*Carassius auratus* (Goldfish)—Cypriniformes; C—*Perca fluviatilis* (European perch—Perciformes; D—*Cottus gobio* (Bullhead)—Perciformes; E—*Clarias gariepinus* (North African catfish)—Siluriformes; F—*Cichla ocellaris* (Peacock cichlid)—Chichliformes. (Fish species numbers in the FishBase classification list, Available online: https://www.fishbase.se/tools/Classification/ClassificationList.php, accessed on 3 February 2025).

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
