# Peer review of "Coccidia (Apicomplexa: Eucoccidiorida) of Freshwater Fish"

_microorganisms, 2025, doi:10.3390/microorganisms13020347_

Round 1

Reviewer 1 Report

Comments and Suggestions for Authors

The authors have compiled reports of coccidian parasites of freshwater fish. It is probably good to generate such lists from time-to-time. The list does seem rather complete, although, the authors never addressed the 59 reports not retrieved. This implies some missing data. And although, not a coccidian, an honorable mention of the haemosporidian Mesnilium malariae in the Introduction might be nice. Misra K.K., Haldar D.P. and Chakravarty M.M. (1972) Observations on Mesnilium malariae gen.nov., spec.nov. (Haemosporidia, Sporozoa) from the fresh water teleost, Ophicephalus punctatus Bloch. Arch. Protistenkd 114, 444-452 

The taxonomy is a bit outdated. For example ref 12 Adl et al was updated in 2019.  https://doi.org/10.1111/jeu.12691 . In addition, the authors should have a look at Votýpka, J., Modrý, D., Oborník, M., Šlapeta, J., Lukeš, J. (2016). Apicomplexa. In: Archibald, J., et al. Handbook of the Protists. Springer, Cham. https://doi.org/10.1007/978-3-319-32669-6_20-1

There is no general agreement on the classification of protozoa including the apicomplexa, but Table 1 should be updated. For example, as the authors acknowledged the position of cryptosporidia is uncertain. However, it should probably be placed at an equal level with Coccidia in Table 1 to make things more accurate. Similarly it doesn't appear the Eucoccidiaorida is still recognized and that column in Table 1 can be removed. And some now consider Conoidasida a subphylum and Coccidia a class. The exact names for the various heirarchies is not extremely important, but getting the heirarchies right would be beneficial. I also suggest making a simple figure or table to show the levels of taxonmy and move the complete Table 1 to an Appendix. 

I also recommend making simple diagrams or tables to summarize tables 3-7 and moving the complete tables to appendices. This would help with the flow of the paper and improve readability. Table 8 can be reduced in size by making the font smaller and less spacing between the rows. 

Simple diagrams showing generic adeleorina and eimeriorina life cycles would also improve the paper and help the readers. 

line 284, remove Cryptosporidium

line 338. elevate this to 3. Cryptosporidium and move to after the eimerids. The authors should also consider that some of the species described as cryptosporidia based on morphology may actually be eimerids. It may be worth mentioning in this section.  

lines 18-21, these two sentences could be substantially reduced by saying something to the effect of '173 apicomplexan parasites in freshwater fish have been described.' As currently written it is too much detail for an abstract and difficult to read. 

Comments on the Quality of English Language

There are some minor language issues

lines 10, 30, change may act as to includes

line 21 review is mispelled

line 22, ...indicate a high prevalence of infection ...

line 39, change much to substantial and delete different

line 40, change are to may be and end the sentence at involved. 

line 48-50 reticuloendothelial system

line 52, six (not sex)

line 74, delete (Fig. 1) [sentence started with Fig. 1]

lines 86, definitive and intermediate hosts can be complicated in the apicomplexa. I recommend just talking about vertebrate and invertebrate hosts and not try to classify the type of host. 

line 137, delete As we informed you, 

line 185, this is not capitalized

line 230, hematomas is probably not the right word here

Author Response

Thank you very much for taking the time to review this manuscript. Please find the detailed responses below and the corresponding revisions highlighted in the re-submitted files

Comments: [The list does seem rather complete, although, the authors never addressed the 59 reports not retrieved. This implies some missing data.]

Response: [Thank you, reviewer. Therefore, 59 reports are the simple and the oldest reports.]

Comments: [And although, not a coccidian, an honorable mention of the haemosporidian Mesnilium malariae in the Introduction might be nice. Misra K.K., Haldar D.P. and Chakravarty M.M. (1972) Observations on Mesnilium malariae gen.nov., spec.nov. (Haemosporidia, Sporozoa) from the fresh water teleost, Ophicephalus punctatus Bloch. Arch. Protistenkd 114, 444-452 

Response [Thank you reviewers.  Haemosporidia are not included in the review].

Comments: [The taxonomy is a bit outdated. For example ref 12 Adl et al was updated in 2019.  https://doi.org/10.1111/jeu.12691 . In addition, the authors should have a look at Votýpka, J., Modrý, D., Oborník, M., Šlapeta, J., Lukeš, J. (2016). Apicomplexa. In: Archibald, J., et al. Handbook of the Protists. Springer, Cham. https://doi.org/10.1007/978-3-319-32669-6_20-1

Response: [ Thank you, reviewer. The taxonomy is also taken from “Schoch, C.L., Ciufo, S., Domrachev, M., Hotton, C.L., Kannan S., Khovanskaya, R., Leipe, D., Mcveigh, R., O'Neill, K., Robbertse, B., Sharma, S., Soussov, V., Sullivan, J.P., Sun, L., Turner, S., Karsch-Mizrachi, I. NCBI Taxonomy: a comprehensive update on curation, resources and tools. Database (Oxford). 2020, baaa062. doi: 10.1093/database/baaa062”. We changed the references – Adl et al, 2012 to Votýpka et al, 2016. ]

Comments: [There is no general agreement on the classification of protozoa including the apicomplexa, but Table 1 should be updated. For example, as the authors acknowledged the position of cryptosporidia is uncertain. However, it should probably be placed at an equal level with Coccidia in Table 1 to make things more accurate. Similarly, it doesn't appear the Eucoccidiaorida is still recognized and that column in Table 1 can be removed. And some now consider Conoidasida a subphylum and Coccidia a class. The exact names for the various heirarchies is not extremely important, but getting the heirarchies right would be beneficial. I also suggest making a simple figure or table to show the levels of taxonmy and move the complete Table 1 to an Appendix.]

Response: [Thank you, reviewer. In Table 1, the classification of apicomplexan is based on Schoch, C.L., Ciufo, S., Domrachev, M., Hotton, C.L., Kannan S., Khovanskaya, R., Leipe, D., Mcveigh, R., O'Neill, K., Robbertse, B., Sharma, S., Soussov, V., Sullivan, J.P., Sun, L., Turner, S., Karsch-Mizrachi, I. NCBI Taxonomy: a comprehensive update on curation, resources and tools. Database (Oxford). 2020, baaa062. doi: 10.1093/database/baaa062.”]

Comments: [I also recommend making simple diagrams or tables to summarize tables 3-7 and moving the complete tables to appendices. This would help with the flow of the paper and improve readability. Table 8 can be reduced in size by making the font smaller and less spacing between the rows. 

Response: [Thank you, reviewer. We agree with this comment. All tables are moved into appendices]

Comments [Simple diagrams showing generic adeleorina and eimeriorina life cycles would also improve the paper and help the readers.] 

Response: [Thank you, reviewer. We agree with this comment. Simple diagrams are added]. In the revised manuscript this change can be found – p.54.

Comments: [line 284, remove Cryptosporidium]

Response: [Thank you, reviewer. We don’t understand this comment—why? Cryptosporidium is an Apicomplexan parasite, and we leave it as it is. Kindly ask to clarify the issue.]

Comments: [line 338. elevate this to 3. Cryptosporidium and move to after the eimerids. The authors should also consider that some of the species described as cryptosporidia based on morphology may actually be eimerids. It may be worth mentioning in this section.]

Response: [Thank you reviewer. We kindly disagree. The position of genus Cryptosporidium on the evolutionary tree of Apicomplexa is unresolved. Cryptosporidium was transferred from Coccidia to class Gregarinomorphea as new subclass Cryptogregaria. We noted this on the text. But we prefere the traditional classification of Cryptosporidium based on Schoch et al, 2020 which the genus Cryptosporidium belong Apicomplexa: Apicomplexa; Conoidasida; Coccidia; Eucoccidiorida; Eimeriorina; Cryptosporidiidae; Cryptosporidium.]

Comments: lines 18-21, these two sentences could be substantially reduced by saying something to the effect of '173 apicomplexan parasites in freshwater fish have been described.' As currently written it is too much detail for an abstract and difficult to read. 

Response: [In the freshwater fish found 173 described species: Cyrilia and Hepatozoon, both with 1; Dactylosoma and Isospora, with 2; Octosporella, with 3; Babesiosoma with 5; Cryptosporidium and Calyptospora, both with 6; Haemogregarina, with 7; Goussia, with 52; and Eimeria, with 88 described species. – In the freshwater fish found 173 described species.]. Thank you for pointing this out. We agree with this comment. In the revised manuscript this change can be found – p. 1; l. 17.

Comments on the Quality of English Language

There are some minor language issues

Comments 1: [lines 10, 30, change may act as to includes]

Response 1: [The Phylum Apicomplexa may act as endoparasites – The Phylum Apicomplexa includes as endoparasites]. Thank you for pointing this out. We agree with this comment. In the revised manuscript this change can be found – p. 1; l. 9, 25.

Comments 2: [line 21 review is misspelled]

Response 2: [This review – This review]. Thank you for pointing this out. We agree with this comment. In the revised manuscript this change can be found – p. 1; l. 17.

Comments 3: [line 22, ...indicate a high prevalence of infection ...]

Response 3: […indicate that infection of high prevalence… – ...indicate a high prevalence of infection…]. Thank you for pointing this out. We agree with this comment. In the revised manuscript this change can be found – p. 1; l. 19.

Comment 4: [line 39, change much to substantial and delete different]

Response 4: […there is much variation among different apicomplexan groups. – …there is substantial variation among apicomplexan groups.]. Thank you for pointing this out. We agree with this comment. In the revised manuscript this change can be found – p.1; l. 35.

Comment 5: [line 40, change are to may be and end the sentence at involved].

Response 5: [Both asexual and sexual reproductions are involved, although some apicomplexans skip one or the other stage. – Both asexual and sexual reproductions may be involved.]. Thank you for pointing this out. We agree with this comment. In the revised manuscript this change can be found – p.1; l. 36.

Comment 6: [line 48-50 reticuloendothelial system]

Response 6: [reticule endothelial cells – reticuloendothelial system]. Thank you for pointing this out. We agree with this comment. In the revised manuscript this change can be found – p. 1; l. 43.

Comment 7: [line 52, six (not sex)]

Response 7: [sex – six]. Thank you for pointing this out. We agree with this comment. In the revised manuscript this change can be found – p.1; l. 46.

Comment 8: [line 74, delete (Fig. 1) [sentence started with Fig. 1]].

Response 8: [Fig. 1 illustrates the selection of studies employing the PRISMA flowchart [11] (Fig.1) – Fig. 1 illustrates the selection of studies employing the PRISMA flowchart [11].]. Thank you for pointing this out. We agree with this comment. In the revised manuscript this change can be found – p. 2; l. 66-67 .

Comment 9: [lines 86, definitive and intermediate hosts can be complicated in the apicomplexa. I recommend just talking about vertebrate and invertebrate hosts and not try to classify the type of host.] 

Response 9: [Parasites of the genus Dactylosoma and Babesiosoma are heteroxenous coccidians that cycle between two hosts: an invertebrate, which is the primary host, and a vertebrate, which is an intermediate host (leech vector) – Parasites of the genus Dactylosoma and Babesiosoma are heteroxenous coccidians that cycle between two hosts: an invertebrate and a vertebrate host.]. Thank you for pointing this out. We agree with this comment. In the revised manuscript this change can be found – p.2; l.78-80.

Comment 10; [line 137, delete As we informed you]

Response 10: [As we informed you, the first described Babesiosoma species – The first described Babesiosoma species…]. Thank you for pointing this out. We agree with this comment. In the revised manuscript this change can be found – p.4; l. 127.

Comment 11: [line 185, this is not capitalized]

Response 11: […This stage is not found – …this stage is not found].  Thank you for pointing this out. We agree with this comment. In the revised manuscript this change can be found – p.4; l. 169

Comment 12: [line 230, hematomas is probably not the right word here]

Response 12: [The genus Cyrilia was created in 1981 for hematomas that infect freshwater fishes and are transmitted by the bite of African fish leeches, Batracobdelloides tricarinata – The genus Cyrilia was created in 1981. Species of Cyrilia infect freshwater fishes and are transmitted by the bite of African fish leeches, Batracobdelloides tricarinata.]. Thank you for pointing this out. We agree with this comment. In the revised manuscript this change can be found – p.5; l.217-212.

Reviewer 2 Report

Comments and Suggestions for Authors

REVIEW REPORT

The review paper aimed to compile all available information about coccidia species found in freshwater fishes. It’s a robust review, containing important information about several parasite genera and species—many of them affecting productivity in aquaculture—including morphometric analysis, host species, site of infection, and global distribution. Although very interesting and complete, there is room for improvement in the review.

General comments

(1) Language revision by a native English speaker is advisable, or the use of AI tools to correct English. There are several truncated phrases and other problems with English writing. Examples: Lines 89-95, 200-203, 246-247, 296-303, and 17-19 (*the line numbering restarted at page 43*).

(2) Authors must check the entire document and correct the spelling of species names, which must always be in italics. The genera should be abbreviated if followed by the species name after the first appearance in the text.

(3) Avoid redundancies and repetition of information throughout the manuscript.

Abstract

(4) Line 11 – replace “affect high productivity” with “affect productivity”

Introduction

(5) Lines 27-29 – avoid redundancies and make the text more concise. Replace “The phylum Apicomplexa Levine et al. 1980 comprises a large group of obligate, intracellular protist parasites. Intracellular single-celled parasites from the large phylum Apicomplexa are the most prevalent and morbidity-causing pathogens worldwide.” with “The phylum Apicomplexa (Levine et al. 1980) comprises a large group of obligate, intracellular protist parasites that are among the most prevalent and morbidity-causing pathogens worldwide.”

(6) Line 32 – replace “affect high productivity” with “affect productivity”

(7) Lines 51-52 – remove the repetitive text “Intracellular blood parasites are a diverse group of adeleorinid coccidia (Apicomplexa: Adeleorina).”

(8) Line 52 – replace “These blood parasites” with “Adeleorina parasites”

(9) Line 52 – replace “among sex families” with “among six families”

(10) Line 56 – replace “genus” with “genera”

(11) Table 1 – replace the header “Main morphologic characteristics” with “Main morphological characteristics”

(12) Lines 59-60 – replace “Historically, however, the term “coccidia” has referred mainly to obligate intracellular protozoa of the genera Eimeria (Schneider, 1875) and Isospora (Schneider, 1881) in the family Eimeriidae.” with “The term “coccidia” has referred mainly to obligate intracellular protozoa of the genera Eimeria (Schneider, 1875) and Isospora (Schneider, 1881) in the family Eimeriidae, suborder Eimeriorina.”

(13) Line 65 – replace “also in different” with “also different.”

(14) Line 69 – the search strategy should be placed in a specific section, such as “Methods” or “Study design,” not in the Introduction section.

(15) Line 69 – is there any reason not to include the Web of Science Core Collection, one of the most widely used academic databases known for its rigorous curation and selection standards?

(16) Line 71 – replace “Coccida” with “Coccidia,” and check if the term was correctly written during the search in databases.

(17) Figure 1 – in the screening process, include the reason for excluding papers at each step.

(18) In Figure 1, authors state they excluded 53 reports with “Old information,” and then they say 151 “New studies” were included in the review. What does this mean? Did authors apply a time restriction? What was it? It’s not clear in the text.

Adeleorina

(19) Lines 84–85 – replace “The genus Dactylosoma and Babesisoma belong to the family Dactylosomatidae [12,13]. Parasites of the genus Dactylosoma and Babesiosoma are heteroxenous coccidians.” with “The genera Dactylosoma and Babesiosoma belong to the family Dactylosomatidae and include heteroxenous coccidians.”

(20) Line 87 – remove “(leech vector)”

(21) Lines 89–95 – the text is confusing. Please rewrite to improve clarity.

(22) Lines 98–100 – replace “In the fish host erythrocyte Babesiosoma differs from Dactylosoma in lacking a nuclear karyosome, possessing vacuolated and only slightly granular cytoplasm, and producing only four merozoites in a rosette or cross.” with “In the fish host erythrocyte, Babesiosoma differs from Dactylosoma by lacking a nuclear karyosome, possessing a vacuolated and only slightly granular cytoplasm, and producing only four merozoites in a rosette or cross formation.”

(23) Line 116 – correct “are are”

(24) Line 129 – replace “and concluded that” with “leading to the conclusion that”

(25) Lines 131–133 – this paragraph should be moved from here to the end of the section and combined with the paragraph in lines 165–170.

(26) Line 137 – remove “As we informed you”

(27) Table 2 – replace “species are currently recognized in freshwater hosts and morphometric parameters of this species.” with “species currently recognized in freshwater hosts, along with their morphometric parameters.”

(28) Line 158 – replace “Babesiosoma sp. they were recorded” with “Babesiosoma sp. was recorded”

(29) Line 171 – replace “Haemogregarina, Cyrilia.” with “Haemogregarina, Cyrilia, and Desseria.”

(30) Line 178 – replace “Haemogregarinacontain” with “Haemogregarina contain”

(31) Line 179 – replace “genus” with “genera”

(32) Line 185 – replace “while in Desseria spp. This stage is not found.” with “while in Desseria spp., this stage is not found.”

(33) Line 194 – replace “haemogregarina bigemina,” with “Haemogregarina bigemina,

(34) Lines 200-203 – the text is confusing; please rewrite

(35) Lines 204-205 – replace “As we noted, Haemogregarina mainly infects marine fishes. Still, some species as Haemogregarina catostomi, H.vltanensis, H. majeedin, H.daviesensis, were recorded in fresh-water fish:” with “Other species that infect fresh-water fish include Haemogregarina catostomi, H. vltanensis, H. majeedin, and H. daviesensis

(36) Lines 222-223 – replace “erythrocytes four specimens” with “erythrocytes of four specimens”

(37) Line 228 – replace “are found” with “are also found”

(38) Line 233 – replace “Cururu stingray” with “Potamotrygon wallacei (Cururu stingray)”

(39) Lines 242-243 – replace “freshwater fish (Potamotrygon wallacei (stingray cururu))” with “Cururu stingray”

(40) Line 243 – replace “Negro River (The Amazonian rivers)” with “Rio Negro, a river of Amazonas, Brazil”

(41) Lines 246-247 – the text is confusing. Please rewrite for clarity

(42) Line 264 – replace “lifecycle” with “life cycle”

Eimeriorina

(43) Lines 283-286 – avoid repetitive information (lines 61-65)

(44) Lines 288-193 – include references to the text

(45) Line 291 – replace “parasitising” with “parasitizing”

(46) Line 295 – “contain” should not be italicized

(47) Line 296 – replace “Calyptospa” with “Calyptospora

(48) Lines 296-303 – the text is truncated. Please rewrite for clarity.

(49) Line 304 – replace “For the first time coccidia in fish was found and described 110 years ago.” with “Coccidia in fish were first discovered and described 110 years ago”

(50) Line 331 – replace “Brasilian” with “Brazilian”

(51) Table 3 – replace “Orders” with “Fish order” in the first column of the table

(52) Lines 365, 366, 376 – remove duplicated words “characterization” and “recognized”

(53) Table 4 – replace “Location” with “Site of infection” in the first line of the table

(54) Table 4 – what do the “Pisgine genotypes,” “Novel genotype,” and “Rat genotype” refer to? Are they different genotypes of Cryptosporidium species? If so, the species names should still be included in this table column (for example, Cryptosporidium parvum, pisgine genotype III)

(55) Line 8 (the line numbering restarted at page 23) – replace “genus” with “genera”

(56) Line 12 – replace “these” with “the”

(57) Lines 13-14 – the authors state that “fish coccidia are currently classified as all belonging to the same genus,” but in the same section, the authors refer to several genera (Eimeria, Goussia, Isospora, Octosporella). Please, correct inconsistencies in the text

(58) Table 5 – replace “Eimeria species are currently recognized in freshwater hosts, and morphometric parameters of this species.” with “Eimeria species currently recognized in freshwater hosts, along with their morphometric parameters”

(59) Table 5 – replace “Orders” with “Fish order” in the first column of the table

(60) Table 5 – replace “Location” with “Site of infection” in the first line of the table

(61) Table 6 – replace “Goussia species currently recognized in freshwater hosts and morphometric parameters of this species.” with “Goussia species currently recognized in freshwater hosts, along with their morphometric parameters”

(62) Table 6 – replace “Location” with “Site of infection” in the first line of the table

(63) Table 7 – replace “Isospora and Octosporella species recognized in freshwater hosts and morphometric parameters of this species.” with “Isospora and Octosporella species currently recognized in freshwater hosts, along with their morphometric parameters”

(64) Table 7 – replace “Location” with “Site of infection” in the first line of the table

Distribution of coccidian freshwater fish

(65) Lines 17-19 (*the line numbering restarted again at page 43*) – replace “Our search resulted in 200 defined and undefined species of adeleid and eimeriid coccidians distributed among 11 genera are found in representatives of only 22 orders of freshwater fish.” with “Our search identified 200 defined and undefined species of adeleid and eimeriid coccidians, distributed among 11 genera, found in representatives of only 22 orders of freshwater fish”

(66) Lines 22-27 – replace “In the freshwater fish found 173 described species: Cyrilia and Hepatozoon with one described species, Dactylosoma and Isospora, both with two described species each; Octosporella, with three described species; Babesiosoma with five described species, Cryptosporidium with six described species (4-specific for piscine host, two specific for mammalia), Calyptospora with six described species, Haemogregarina with seven described species, Goussia, with 52 species; and Eimeria, with 88 species (Table 2, 3, 4, 5, 6, 7).” with “In freshwater fish, 173 described species have been identified: Cyrilia and Hepatozoon, each with one described species; Dactylosoma and Isospora, both with two described species; Octosporella, with three described species; Babesiosoma, with five described species; Cryptosporidium, with six described species (four specific to piscine hosts and two specific to mammals); Calyptospora, with six described species; Haemogregarina, with seven described species; Goussia, with 52 described species; and Eimeria, with 88 described species (Tables 2-7)”

(67) Lines 28-29 – replace “For some species, it was only possible to obtain also information concerning the characteristics considered for this review.” with “For some species, it was only possible to obtain information regarding the characteristics considered in this review”

(68) Table 8 – replace “Coccida” with “Coccidia” in the first line of the table

(69) Table 10 – revise the table, since there are repetitive lines

(70) Page 50, paragraph 1 (*at this time, no line numbering was included in lines at all*) – replace “continental” with “continent” in the subtitle

(71) Page 50, paragraph 1 – replace “Hamogregarine” with “Haemogregarine

(72) In the subsequent paragraphs, authors should correct the spelling of the words “freshwater” (repeated four times), “spread,” and “Amazon”

Conclusion

(73) This section should be rewritten to include the most important information obtained from the data retrieved from the papers included in the review. It should not merely repeat the results. Also, it should include future directions for research being conducted in this research area, especially concerning the issue of low productivity in aquaculture in which fish are infected by those parasites.

(74) I also recommend the inclusion, in this section, of a figure with representative images of fishes of each order and the main genera of parasites found in them. This would greatly improve a broad understanding of all data compiled in the review.

Comments on the Quality of English Language

Please, see "General comments".

Author Response

Thank you very much for taking the time to review this manuscript. Please find the detailed responses below and the corresponding revisions highlighted in the re-submitted files.

General comments

Comments 1: [Language revision by a native English speaker is advisable, or the use of AI tools to correct English. There are several truncated phrases and other problems with English writing. Examples: Lines 89-95, 200-203, 246-247, 296-303, and 17-19 (*the line numbering restarted at page 43*)].

Response 1: [Thank you for pointing this out. We agree with this comment. We edited the whole MS again.]

Comments 2: [Authors must check the entire document and correct the spelling of species names, which must always be in italics. The genera should be abbreviated if followed by the species name after the first appearance in the text.]

Response 2: [Thank you for pointing this out. We agree with this comment and the changes have been done]

Comments 3: [Avoid redundancies and repetition of information throughout the manuscript.]

Response 3: [Thank you for pointing this out. We agree with this comment.]

Abstract

Comment 4: [Line 11 – replace “affect high productivity” with “affect productivity”]

Response 4: […affect high productivity… – … affect productivity…]. Thank you for pointing this out. We agree with this comment. In the revised manuscript this change can be found – p.1; l.10.

Introduction

Comment 5: [Lines 27-29 – avoid redundancies and make the text more concise. Replace “The phylum Apicomplexa Levine et al. 1980 comprises a large group of obligate, intracellular protist parasites. Intracellular single-celled parasites from the large phylum Apicomplexa are the most prevalent and morbidity-causing pathogens worldwide.” with “The phylum Apicomplexa (Levine et al. 1980) comprises a large group of obligate, intracellular protist parasites that are among the most prevalent and morbidity-causing pathogens worldwide.”]

Response 5: [The phylum Apicomplexa Levine et al. 1980 comprises a large group of obligate, intracellular protist parasites. Intracellular single-celled parasites from the large phylum Apicomplexa are the most prevalent and morbidity-causing pathogens worldwide. – The phylum Apicomplexa (Levine et al. 1980) comprises a large group of obligate, intracellular protist parasites among the most prevalent and morbidity-causing pathogens worldwide]. Thank you for pointing this out. We agree with this comment. In the revised manuscript this change can be found – p.1; l. 23-25.

Comment 6: [Line 32 – replace “affect high productivity” with “affect productivity”]

Response 6: […affect high productivity… – … affect productivity…]. Thank you for pointing this out. We agree with this comment. In the revised manuscript this change can be found – p.1; l.27.

Comment 7: [Lines 51-52 – remove the repetitive text “Intracellular blood parasites are a diverse group of adeleorinid coccidia (Apicomplexa: Adeleorina).”]

Response 7: [Intracellular blood parasites are a diverse group of adeleorinid coccidia (Apicomplexa: Adeleorina). These blood parasites are currently distributed among six families.  – Adeleorina parasites are currently distributed among six families.]. Thank you for pointing this out. We agree with this comment. In the revised manuscript this change can be found – p.1; l.46.

Comment 8: [Line 52 – replace “These blood parasites” with “Adeleorina parasites”]

Response 8: [These blood parasites … – Adeleorina parasites …]. Thank you for pointing this out. We agree with this comment. In the revised manuscript this change can be found – p.1; l.46.

Comment 9: [Line 52 – replace “among sex families” with “among six families”]

Response 9: […among sex families – … among six families]. Thank you for pointing this out. We agree with this comment. In the revised manuscript this change can be found – p.1; l.46.

Comment 10: [Line 56 – replace “genus” with “genera”]

Response 10: […genus …– … genera…]. Thank you for pointing this out. We agree with this comment. In the revised manuscript this change can be found – p.1; l. 50.

Comment 11: [Table 1 – replace the header “Main morphologic characteristics” with “Main morphological characteristics”]

Response 11: [Main morphologic characteristics …– Main morphological characteristics …]. Thank you for pointing this out. We agree with this comment. In the revised manuscript this change can be found – p. 21;

Comment 12: [Lines 59-60 – replace “Historically, however, the term “coccidia” has referred mainly to obligate intracellular protozoa of the genera Eimeria (Schneider, 1875) and Isospora (Schneider, 1881) in the family Eimeriidae.” with “The term “coccidia” has referred mainly to obligate intracellular protozoa of the genera Eimeria (Schneider, 1875) and Isospora (Schneider, 1881) in the family Eimeriidae, suborder Eimeriorina.”]

Response 12: [Historically, however, the term “coccidia” has referred mainly to obligate intracellular protozoa of the genera Eimeria (Schneider, 1875) and Isospora (Schneider, 1881) in the family Eimeriidae. – The term “coccidia” has referred mainly to obligate intracellular protozoa of the genera Eimeria (Schneider, 1875) and Isospora (Schneider, 1881) in the family Eimeriidae, suborder Eimeriorina.]. Thank you for pointing this out. We agree with this comment. In the revised manuscript this change can be found – p.1; l.50-52.

Comment 13: [Line 65 – replace “also in different” with “also different.”]

Response 13: […also in different …– … also different …]. Thank you for pointing this out. We agree with this comment. In the revised manuscript this change can be found – p.2; l.56.

Comment 14: [Line 69 – the search strategy should be placed in a specific section, such as “Methods” or “Study design” not in the Introduction section.].

Response 14: [I added the section “Study design”.]  Thank you for pointing this out. We agree with this comment. In the revised manuscript this change can be found – p.2; l. 61.

Comment 15: [Line 69 – is there any reason not to include the Web of Science Core Collection, one of the most widely used academic databases known for its rigorous curation and selection standards?]

Response 15: [Thank you, reviewer. Almost all of the articles of Web of Science Core Collection databases we can find in SCOPUS, Pub Med and Google Scholar.]

Comments 16: [Line 71 – replace “Coccida” with “Coccidia,” and check if the term was correctly written during the search in databases.]

Response 16: […Coccida …– … Coccidia …]. Thank you for pointing this out. We agree with this comment. In the revised manuscript this change can be found – p.2; l.64.

Comment 17: [Figure 1 – in the screening process, include the reason for excluding papers at each step].

Response 17: [Was done]. Thank you for pointing this out. We agree with this comment. In the revised manuscript this change can be found – p.2.

Comment 18: [In Figure 1, authors state they excluded 53 reports with “Old information,” and then they say 151 “New studies” were included in the review. What does this mean? Did authors apply a time restriction? What was it? It’s not clear in the text.]

Response 18: ["New studies" must not be here, was corrected]. Thank you for pointing this out. We I agree with this comment. In the revised manuscript this change can be found – p.2.

 Adeleorina

Comment 19: [Lines 84–85 – replace “The genus Dactylosoma and Babesisoma belong to the family Dactylosomatidae [12,13]. Parasites of the genus Dactylosoma and Babesiosoma are heteroxenous coccidians.” with “The genera Dactylosoma and Babesiosoma belong to the family Dactylosomatidae and include heteroxenous coccidians.”]

Response 19: The genus Dactylosoma and Babesisoma belong to the family Dactylosomatidae [12,13]. Parasites of the genus Dactylosoma and Babesiosoma are heteroxenous coccidians that cycle between two hosts: an invertebrate, which is the primary host, and a vertebrate, which is an intermediate host (leech vector) [14]. – The genera Dactylosoma and Babesiosoma belong to the family Dactylosomatidae and include heteroxenous coccidians that cycle between two hosts: an invertebrate and a vertebrate host [12, 13, 14]. Thank you for pointing this out. We agree with this comment. In the revised manuscript this change can be found – p.3; l.77-79.

Comment 20: [Line 87 – remove “(leech vector)”]

Response 20: [“(leech vector)” was removed]. Thank you for pointing this out. We agree with this comment. In the revised manuscript this change can be found – p.2.

Comment 21: [Lines 89–95 – the text is confusing. Please rewrite to improve clarity.]

Response 21: [Members of the family Dactylosomatidae have remained, until recently, some of the most poorly understood members of the species Apicomplexa despite the long period that has elapsed since their discovery. Have clarified the relationships between the dactylosomatid genera Dactylosoma and Babesiosoma and have more clearly defined the taxonomic status of species in the family Dactylosomatidae and their relationships with other Apicomplexa. Species of the family Dactylosomatidae are produced in erythrocytes of cold-blooded vertebrates 4 to 16 merozoites; Species of the genus Babesiosoma, which made only four merozoites, species of the genus Dactylosoma more than four merozoites during intra-erythrocytic replication [6, 15] – Members of the family Dactylosomatidae have remained, until recently, some of the most poorly understood members of the species Apicomplexa despite the long period that has elapsed since their discovery. We have clarified the relationships between the dactylosomatid genera Dactylosoma and Babesiosoma. Species of the genus Babesiosoma are produced in erythrocytes of cold-blooded vertebrates four merozoites, and species of the genus Dactylosoma more than four merozoites during intra-erythrocytic replication [6, 15]]. Thank you for pointing this out. We agree with this comment. In the revised manuscript this change can be found – p.3; l.80-86.

Comment 22: [Lines 98–100 – replace “In the fish host erythrocyte Babesiosoma differs from Dactylosoma in lacking a nuclear karyosome, possessing vacuolated and only slightly granular cytoplasm, and producing only four merozoites in a rosette or cross. – In the fish host erythrocyte, Babesiosoma differs from Dactylosoma by lacking a nuclear karyosome, possessing a vacuolated and only slightly granular cytoplasm, and producing only four merozoites in a rosette or cross formation.]

Response 22: [The fish host erythrocyte Babesiosoma differs from Dactylosoma in lacking a nuclear karyosome, possessing vacuolated and only slightly granular cytoplasm, and producing only four merozoites in a rosette or cross. – The fish host erythrocyte, Babesiosoma differs from Dactylosoma by lacking a nuclear karyosome, possessing a vacuolated and only slightly granular cytoplasm, and producing only four merozoites in a rosette or cross formation]. Thank you for pointing this out. We agree with this comment. In the revised manuscript this change can be found – p.3; l. 88-90.

Comment 23: [Line 116 – correct “are are”]

Response 23: […are are …– … are …]. Thank you for pointing this out. We agree with this comment. In the revised manuscript this change can be found – p.3; l.106.

Comments 24: Line 129 – replace “and concluded that” with “leading to the conclusion that”

Response 24: […and concluded that …– … leading to the conclusion that …]. Thank you for pointing this out. We agree with this comment. In the revised manuscript this change can be found – p.3; l. 119.

Comments 25: Lines 131–133 – this paragraph should be moved from here to the end of the section and combined with the paragraph in lines 165–170.

Response 25: [In conclusion, currently, there are five recognized species of Dactylosoma, two of which infect fish hosts, namely Dactylosoma lethrinorum Saunders, 1960 and Dactylosoma salvelini Fantham, Porter and Richardson, 1942. (table 2) [25] – There are five recognized species of Dactylosoma, two of which infect fish hosts, namely Dactylosoma lethrinorum Saunders, 1960 and Dactylosoma salvelini Fantham, Porter and Richardson, 1942. (table 2) [25]]. Thank you for pointing this out. We agree with this comment. We couldn’t remove it from here to the end of the section because of the citation [25] but I deleted “in conclusion”]. In the revised manuscript this change can be found – p.3; l.120-122.

Comments 26: Line 137 – remove “As we informed you”

Response 26: [As we informed you, the first described Babesiosoma species – The first described Babesiosoma species…]. Thank you for pointing this out. We agree with this comment. In the revised manuscript this change can be found – p.4; l.127.

Comments 27: [Table 2 – replace “species are currently recognized in freshwater hosts and morphometric parameters of this species.” with “species currently recognized in freshwater hosts, along with their morphometric parameters.”]

Response 27: [species are currently recognized in freshwater hosts and morphometric parameters of this species. – species currently recognized in freshwater hosts, along with their morphometric parameters]. Thank you for pointing this out. We agree with this comment. In the revised manuscript this change can be found – p.23.

Comments 28: [Line 158 – replace “Babesiosoma sp. they were recorded” with “Babesiosoma sp. was recorded”]

Response 28: [Babesiosoma sp. they were recorded… – Babesiosoma sp. was recorded …]. Thank you for pointing this out. We agree with this comment. In the revised manuscript this change can be found – p.4; l.143.

Comments 29: [Line 171 – replace “Haemogregarina, Cyrilia.” with “Haemogregarina, Cyrilia, and Desseria.”]

Response 29: [Haemogregarina, Cyrilia … – Haemogregarina, Cyrilia, and Desseria …]. Thank you for pointing this out. We agree with this comment. In the revised manuscript this change can be found – p.4; l.155.

Comments 30: [Line 178 – replace “Haemogregarinacontain” with “Haemogregarina contain”]

Response 30: [Haemogregarinacontain … – Haemogregarina contain …]. Thank you for pointing this out. We agree with this comment. In the revised manuscript this change can be found – p.4; l.162.

Comments 31: [Line 179 – replace “genus” with “genera”]

Response 31: […genus … – … genera …]. Thank you for pointing this out. We agree with this comment. In the revised manuscript this change can be found – p.4; l.163.

Comment 32: [Line 185 – replace “while in Desseria spp. This stage is not found.” with “while in Desseria spp., this stage is not found.”]

Response 32: […while in Desseria spp. This stage is not found. – while in Desseria spp., this stage is not found.]. Thank you for pointing this out. We agree with this comment. In the revised manuscript this change can be found – p.4; l. 169.

Comments 33: [Line 194 – replace “haemogregarina bigemina,” with “Haemogregarina bigemina,”]

Response 33: […haemogregarina bigemina... – Haemogregarina bigemina...]. Thank you for pointing this out. We agree with this comment. In the revised manuscript this change can be found – p.5; l.178.

Comments 34: [Lines 200-203 – the text is confusing; please rewrite]

Response 34: [Described 23 Haemogregarina species of freshwater fish, but, many of the remaining species may eventually prove to be Haemogregarina, while others may be better placed in Cyrilia, Hepatozoon, or perhaps other genera [5] – There were 23 Haemogregarina species described of freshwater fish.  But some of them may be placed in Cyrilia, Hepatozoon, or perhaps other genera [5].] Thank you for pointing this out. We agree with this comment. In the revised manuscript this change can be found – p.5; l.184-186.

Comments 35: [Lines 204-205 – replace “As we noted, Haemogregarina mainly infects marine fishes. Still, some species as Haemogregarina catostomi, H.vltanensis, H. majeedin, H.daviesensis, were recorded in fresh-water fish:” with “Other species that infect fresh-water fish include Haemogregarina catostomi, H. vltanensis, H. majeedin, and H. daviesensis”]

Response 35: [As we noted, Haemogregarina mainly infects marine fishes. Still, some species as Haemogregarina catostomi, H.vltanensis, H.majeedin, H.daviesensis, were recorded in fresh-water fish: – Other species that infect fresh-water fish including Haemogregarina catostomi, H.vltanensis, H.majeedin, and H.daviesensis]. Thank you for pointing this out. We agree with this comment. In the revised manuscript this change can be found – p.5; l.187-188.

Comments 36: [Lines 222-223 – replace “erythrocytes four specimens” with “erythrocytes of four specimens”]

Response 36: […erythrocytes four specimens... – erythrocytes of four specimens...]. Thank you for pointing this out. We agree with this comment. In the revised manuscript this change can be found – p.5; l.204-205.

Comments 37: [Line 228 – replace “are found” with “are also found”]

Response 37: […are found... – are also found...]. Thank you for pointing this out. We agree with this comment. In the revised manuscript this change can be found – p.5; l.210.

Comments 38: [Line 233 – replace “Cururu stingray” with “Potamotrygon wallacei (Cururu stingray)”]

Response 38: […Cururu stingray... – Potamotrygon wallacei (Cururu stingray)...]. Thank you for pointing this out. We agree with this comment. In the revised manuscript this change can be found – p.5; l.215.

Comments 39: Lines 242-243 – replace “freshwater fish (Potamotrygon wallacei (stingray cururu))” with “Cururu stingray”

Response 39: […freshwater fish (Potamotrygon wallacei (stingray cururu))... – …Cururu stingray...]. Thank you for pointing this out. We agree with this comment. In the revised manuscript this change can be found – p.5; l. 225.

Comments 40: Line 243 – replace “Negro River (The Amazonian rivers)” with “Rio Negro, a river of Amazonas, Brazil”

Response 40: […Negro River (The Amazonian rivers). –… Rio Negro, a river of Amazonas, Brazil.]. Thank you for pointing this out. We agree with this comment. In the revised manuscript this change can be found – p.5; l.225.

Comments 41: [Lines 246-247 – the text is confusing. Please rewrite for clarity]

Response 41: [Although, some Desseria species are now known to belong to other genera [7,56,57] – Although, some early known Desseria species now belong to other genera [7,56,57]. Thank you for pointing this out. We agree with this comment. In the revised manuscript this change can be found – p.5; l.229.

Comments 42: [Line 264 – replace “lifecycle” with “life cycle”]

Response 42: […lifecycle…– … life cycle…]. Thank you for pointing this out. We agree with this comment. In the revised manuscript this change can be found – p.6; l.247.

Eimeriorina

Comments 43: [Lines 283-286 – avoid repetitive information (lines 61-65)]

Response 43: [A diverse array of eight genera of Apicomplexa parasites (Calyptospora, Cryptosporidium, Eimeria, Goussia, Isospora, Epieimeria, Crystallospora, and Octosporella) have been reported to date in fish, with infections occurring in both marine and freshwater fish species. Two genera, Eimeria and Goussia, are represented by many species associated with freshwater fish [10]. – A diverse array of eight genera of Apicomplexa parasites have been reported in fish, with infections occurring in marine and freshwater fish species. Two genera, Eimeria and Goussia, are represented by many species associated with freshwater fish [10]]. Thank you for pointing this out. We agree with this comment. In the revised manuscript this change can be found – p.6; l. 266-269.

Comments 44: [Lines 288-193 – include references to the text]

Response 44: [Was included references to the text]. Thank you for pointing this out. We agree with this comment. In the revised manuscript this change can be found – p.6; l.276.

Comments 45: [Line 291 – replace “parasitising” with “parasitizing”]

Response 45: […parasitising …– … parasitizing …]. Thank you for pointing this out. We agree with this comment. In the revised manuscript this change can be found – p.6; l.274.

Comments 46: [Line 295 – “contain” should not be italicized]

Response 46: […contain …– … contain …]. Thank you for pointing this out. We agree with this comment.

Comments 47: [Line 296 – replace “Calyptospa” with “Calyptospora”]

Response 47: […Calyptospa …– … Calyptospora …]. Thank you for pointing this out. We agree with this comment. In the revised manuscript this change can be found – p.7; l.286.

Comments 48: [Lines 296-303 – the text is truncated. Please rewrite for clarity.]

Response 48: [We deleted all this text: “Oocysts of Cryptosporidium contain four "naked" sporozoites without sporocysts.  Calyptospora oocysts with four sporocysts, each with two sporozoites; sporocysts with sporopodia,  Eimeria oocysts with four sporocysts, each with two sporozoites; merogony intracellular; Goussia oocysts with four sporocysts, each with two sporozoites; dehiscence suture longitudinal, Epieimeria oocysts with four sporocysts, each with two sporozoites, merogony and gamogony extracellular; sporogony intracellular, Isospora oocysts with two sporocysts, each with four sporozoites,  Crystallospora oocysts with four sporocysts, each with two sporozoites; sporocysts resemble crystals and Octosporella oocysts with eight sporocysts, each with two sporozoites”  and added “Table 1” at the end of the upcoming sentence – The structure of oocysts and sporozoites is of paramount importance for the taxonomy of these parasites (Table 1) [6]. ]. Thank you for pointing this out. We agree with this comment. In the revised manuscript this change can be found – p.6; l. 277-278.  

Comments 49: [Line 304 – replace “For the first time coccidia in fish was found and described 110 years ago.” with “Coccidia in fish were first discovered and described 110 years ago”]

Response 49: [For the first time coccidia in fish was found and described 110 years ago. – Coccidia in fish were first discovered and described 110 years ago]. Thank you for pointing this out. We agree with this comment. In the revised manuscript this change can be found – p.6; l.279.

Comments 50: [Line 331 – replace “Brasilian” with “Brazilian”]

Response 50: […Brasilian …– … Brazilian …]. Thank you for pointing this out. We agree with this comment. In the revised manuscript this change can be found – p.7; l.306.

Comments 51:  [Table 3 – replace “Orders” with “Fish order” in the first column of the table]

Response 51: […Orders …– … Fish order …]. Thank you for pointing this out. We agree with this comment. In the revised manuscript this change can be found – p.26.

Comments 52: [Lines 365, 366, 376 – remove duplicated words “characterization” and “recognized”]

Response 52: [Was done]. Thank you for pointing this out. We agree with this comment. In the revised manuscript this change can be found – p.8; l.342,343, 353.

Comments 53: [Table 4 – replace “Location” with “Site of infection” in the first line of the table]

Response 53: […Location …– … Site of infection …]. Thank you for pointing this out. We agree with this comment. In the revised manuscript this change can be found – p. 27.

Comments 54: [Table 4 – what do the “Pisgine genotypes,” “Novel genotype,” and “Rat genotype” refer to? Are they different genotypes of Cryptosporidium species? If so, the species names should still be included in this table column (for example, Cryptosporidium parvum, piscine genotype III)]

Response 54: [We changed the name of the table: “Cryptosporidium species and genotypes currently recognized in freshwater hosts and morphometric parameters of this species". All genotypes are written like this in all another report: See Golomazou and Karanis, 2020; Golomazou et al., 2021]. Thank you for pointing this out. We agree with this comment. In the revised manuscript this change can be found – p.30.

Comments 55: [Line 8 (the line numbering restarted at page 23) – replace “genus” with “genera”]

Response 55: […genus …– … genera …]. Thank you for pointing this out. We agree with this comment. In the revised manuscript this change can be found – p.8; l.368.

Comments 56: [Line 12 – replace “these” with “the”]

Response 56: […these …– … the …]. Thank you for pointing this out. We agree with this comment. In the revised manuscript this change can be found – p.8; l.372.

Comment 56: [Lines 13-14 – the authors state that “fish coccidia are currently classified as all belonging to the same genus,” but in the same section, the authors refer to several genera (Eimeria, Goussia, Isospora, Octosporella). Please, correct inconsistencies in the text]

Response 57: [Fish coccidia, currently classified as all belonging to the same genus, differ from typical Eimeria species from higher vertebrates in having, as a rule, a thin oocyst wall and en-dogenous sporulation – Fish Eimeria species differ from typical Eimeria species from higher vertebrates in having, as a rule, a thin oocyst wall and endogenous sporulation]. Thank you for pointing this out. We agree with this comment. In the revised manuscript this change can be found – p.8; l.374-375.

Comments 58: [Table 5 – replace “Eimeria species are currently recognized in freshwater hosts, and morphometric parameters of this species.” with “Eimeria species currently recognized in freshwater hosts, along with their morphometric parameters”]

Response 58: [Eimeria species are currently recognized in freshwater hosts, and morphometric parameters of this species. – Eimeria species are recognized in freshwater hosts, and their morphometric parameters]. Thank you for pointing this out. We agree with this comment. In the revised manuscript this change can be found – p.31.

Comments 59: [Table 5 – replace “Orders” with “Fish order” in the first column of the table]

Response 59: […Orders …– … Fish order …]. Thank you for pointing this out. We agree with this comment. In the revised manuscript this change can be found – p.31.

Comments 60: [Table 5 – replace “Location” with “Site of infection” in the first line of the table]

Response 60: […Location …– … Site of infection …]. Thank you for pointing this out. We agree with this comment. In the revised manuscript this change can be found – p.31.

Comments 61: [Table 6 – replace “Goussia species currently recognized in freshwater hosts and morphometric parameters of this species.” with “Goussia species currently recognized in freshwater hosts, along with their morphometric parameters”]

Response 61: Goussia species currently recognized in freshwater hosts and morphometric parameters of this species – Goussia species currently recognized in freshwater hosts, along with their morphometric parameters]. Thank you for pointing this out. We agree with this comment. In the revised manuscript this change can be found – p.40.

Comments 62: [Table 6 – replace “Location” with “Site of infection” in the first line of the table]

Response 62: […Location …– … Site of infection …]. Thank you for pointing this out. We agree with this comment. In the revised manuscript this change can be found – p.40.

Comments 63: [Table 7 – replace “Isospora and Octosporella species recognized in freshwater hosts and morphometric parameters of this species.” with “Isospora and Octosporella species currently recognized in freshwater hosts, along with their morphometric parameters”]

Response 63: [Isospora and Octosporella species recognized in freshwater hosts and morphometric parameters of this species – Isospora and Octosporella species currently recognized in freshwater hosts, along with their morphometric parameters]. Thank you for pointing this out. We agree with this comment. In the revised manuscript this change can be found – p.47.

Comments 64: [Table 7 – replace “Location” with “Site of infection” in the first line of the table]

Response 64: […Location …– … Site of infection …]. Thank you for pointing this out. We agree with this comment. In the revised manuscript this change can be found – p.47.

Distribution of coccidian freshwater fish

Comments 65: [Lines 17-19 (*the line numbering restarted again at page 43*) – replace “Our search resulted in 200 defined and undefined species of adeleid and eimeriid coccidians distributed among 11 genera are found in representatives of only 22 orders of freshwater fish.” with “Our search identified 200 defined and undefined species of adeleid and eimeriid coccidians, distributed among 11 genera, found in representatives of only 22 orders of freshwater fish”]

Response 65: Our search resulted in 200 defined and undefined species of adeleid and eimeriid coccidians distributed among 11 genera found in representatives of only 22 orders of freshwater fish. – Our search identified 200 defined and undefined species of adeleid and eimeriid coccidians, distributed among 11 genera, found in representatives of only 22 orders of freshwater fish]. Thank you for pointing this out. We agree with this comment. In the revised manuscript this change can be found – p.9; l.402-404.

Comments 66: [Lines 22-27 – replace “In the freshwater fish found 173 described species: Cyrilia and Hepatozoon with one described species, Dactylosoma and Isospora, both with two described species each; Octosporella, with three described species; Babesiosoma with five described species, Cryptosporidium with six described species (4-specific for piscine host, two specific for mammalia), Calyptospora with six described species, Haemogregarina with seven described species, Goussia, with 52 species; and Eimeria, with 88 species (Table 2, 3, 4, 5, 6, 7).” with “In freshwater fish, 173 described species have been identified: Cyrilia and Hepatozoon, each with one described species; Dactylosoma and Isospora, both with two described species; Octosporella, with three described species; Babesiosoma, with five described species; Cryptosporidium, with six described species (four specific to piscine hosts and two specific to mammals); Calyptospora, with six described species; Haemogregarina, with seven described species; Goussia, with 52 described species; and Eimeria, with 88 described species (Tables 2-7)”]

Response 66: [In the freshwater fish found 173 described species: Cyrilia and Hepatozoon with one described species, Dactylosoma and Isospora, both with two described species each; Octosporella, with three described species; Babesiosoma with five described species, Cryptosporidium with six described species (4-specific for piscine host, two specific for mammalia), Calyptospora with six described species, Haemogregarina with seven described species, Goussia, with 52 species; and Eimeria, with 88 species (Table 2, 3, 4, 5, 6, 7).  – In freshwater fish, 173 described species have been identified: Cyrilia and Hepatozoon, each with one described species; Dactylosoma and Isospora, both with two described species; Octosporella, with three described species; Babesiosoma, with five described species; Cryptosporidium, with six described species (four specific to piscine hosts and two specific to mammals); Calyptospora, with six described species; Haemogregarina, with seven described species; Goussia, with 52 described species; and Eimeria, with 88 described species (Tables 2-7)]. Thank you for pointing this out. We agree with this comment. In the revised manuscript this change can be found – p.9; l.407-413

Comments 67: [Lines 28-29 – replace “For some species, it was only possible to obtain also information concerning the characteristics considered for this review.” with “For some species, it was only possible to obtain information regarding the characteristics considered in this review”]

Response 67: [For some species, it was only possible to obtain also information concerning the characteristics considered for this review – For some species, it was only possible to obtain information regarding the characteristics considered in this review]. Thank you for pointing this out. We agree with this comment. In the revised manuscript this change can be found – p9; l. 414-415.

Comments 68: [Table 8 – replace “Coccida” with “Coccidia” in the first line of the table]

Response 68: […Coccida …– … Coccidia …]. Thank you for pointing this out. We agree with this comment. In the revised manuscript this change can be found – p.48.

Comments 69: [Table 10 – revise the table, since there are repetitive lines]

Response 69: [Was done] Thank you for pointing this out. We agree with this comment. In the revised manuscript this change can be found – p.51.

Comments 70: [Page 50, paragraph 1 (*at this time, no line numbering was included in lines at all*) – replace “continental” with “continent” in the subtitle]

Response 70: […continental. – … continent.]. Thank you for pointing this out. We agree with this comment. In the revised manuscript this change can be found – p.10; l.456.

Comments 71: Page 50, paragraph 1 – replace “Hamogregarine” with “Haemogregarine”

Response 71: […Hamogregarine. – … Haemogregarine.]. Thank you for pointing this out. We agree with this comment. In the revised manuscript this change can be found – p.10; l.459.

Comments 72: [In the subsequent paragraphs, authors should correct the spelling of the words “freshwater” (repeated four times), “spread,” and “Amazon”]

Response 72: [Was corrected] Thank you for pointing this out. We agree with this comment. In the revised manuscript this change can be found – p.10; l.462, 464, 469, 471, 475, 476.

Conclusion

Comments 73: [This section should be rewritten to include the most important information obtained from the data retrieved from the papers included in the review. It should not merely repeat the results. Also, it should include future directions for research being conducted in this research area, especially concerning the issue of low productivity in aquaculture in which fish are infected by those parasites.]

Response 73: [Was corrected] Thank you for pointing this out. We agree with this comment. In the revised manuscript this change can be found – p.11; l. 480-490.

Comments 74: [I also recommend the inclusion, in this section, of a figure with representative images of fishes of each order and the main genera of parasites found in them. This would greatly improve a broad understanding of all data compiled in the review.]

Response 74: [Thank you, reviewer. We agree with this comment. Images are added]. In the revised manuscript this change can be found – p.55.

Comments on the Quality of English Language

Please, see "General comments".

Round 2

Reviewer 1 Report

Comments and Suggestions for Authors

The changes have greatly improved  the manuscript and it is ready for publication. 

Author Response

Dear Reviewer, thank you for the great editing job on the Manuscript.

Reviewer 2 Report

Comments and Suggestions for Authors

Several issues I pointed out in the first revision still remain, and the authors need to address these issues before the manuscript can be considered for publication.

General

(1) There are still several truncated phrases and other problems with English writing. Examples: Lines 83–90, 184–186, 227–234, 414–415. Please revise the language throughout the entire manuscript.

(2) As pointed out before, the authors must check the entire document and correct the spelling of species names. There are still problems with species naming. Examples: Lines 185, 186, 210.

Abstract

(3) Line 9 – Replace “and these parasitic infections can adversely” with “causing parasitic infections that can adversely.”

Introduction

(4) Line 25 – Avoid redundancies. Replace “The phylum Apicomplexa includes endoparasites” with “It includes endoparasites.”

(5) Line 56 – Replace “not only in fish but also in different groups of vertebrates” with “not only fish but also different groups of vertebrates.”

(6) Line 61 – Include/correct the numbering of all section headings. Example: 1. Introduction, 2. Study design, 3. Adeleorina, 3.1 Dactylosoma and Babesiosoma, and so on.

(7) Line 70, Figure 1 – As already pointed out in the first round of review, in the screening process, the authors must include the reason for excluding papers at each step. The authors state they excluded reports with “Old information,” “Poor quality,” and “Lack of data.” What does this mean? Did the authors apply a time restriction? What was it? What parameters were used to evaluate article quality? What parameters were used to evaluate a lack of data? All this information should be included in detail in the Study Design section. Also, in the figure, replace “SCOPUS” with “Scopus” and “Pub Med” with “PubMed.”

Adeleorina

(8) Line 79 – Replace all occurrences of “Schemes” and “Images” with “Figures,” in sequential order as they appear in the document. Examples: Figure 1, Figure 2, and so on.

(9) Line 81 – Replace “the species Apicomplexa” with “the phylum Apicomplexa.”

(10) Lines 83 to 90 – The text is truncated. Rewrite to make sense.

(11) Line 119 – Replace “concluding that” with “leading to the conclusion that.”

(12) Lines 120 to 122 – As already pointed out, this paragraph should be moved from here to the end of the section and combined with the paragraph in Lines 150–154 to avoid repetition and redundancies. References should be renumbered, if necessary.

(13) Lines 184 to 186 – The sentences are truncated and confusing. Rewrite for clarity.

(14) Lines 227 to 234 – The sentences are truncated and confusing. Rewrite for clarity.

Eimeriorina

(15) Lines 263 to 276 – The sentences are repetitive (with Lines 50 to 56) and/or confusing. Rewrite for conciseness.

Distribution of coccidian freshwater fish

(16) Lines 414 and 415 – The sentence is confusing, as pointed out before. Rewrite for clarity: “For some species, it was only possible to obtain information regarding the characteristics considered in this review.”

Conclusion

(17) The inclusion of fish pictures in Image 1 was nice. Now, the authors should highlight, in the same image, the main genera of parasites found in each group of fish. This would greatly improve the understanding of the data compiled in the review.

Comments on the Quality of English Language

As noted in the first round of revisions, the manuscript has significant language issues that were not fully addressed in this new version.

The authors should consider having the language revised by a native English speaker or using AI tools for English correction. Several truncated sentences and other issues with English writing remain.

Author Response

Thank you very much for taking the time to review this manuscript. Please find the detailed responses below and the corresponding revisions highlighted in the re-submitted files

General

Comment 1: [There are still several truncated phrases and other problems with English writing. Examples: Lines 83–90, 184–186, 227–234, 414–415. Please revise the language throughout the entire manuscript.]

Response 1: [Thank you. We edited the whole MS again.]

Comment 2: [As pointed out before, the authors must check the entire document and correct the spelling of species names. There are still problems with species naming. Examples: Lines 185, 186, 210.]

Response 2: […Cyrilia, Hepatozoon…– …Cyrilia, Hepatozoon…, …sand sculpin Leocottus Kessler… – …Kessler's sculpin Leocottus kesslerii…]. Thank you. In the revised manuscript this change can be found – p. 5; l. 216-217.

Abstract

Comment 3: [Line 9 – Replace “and these parasitic infections can adversely” with “causing parasitic infections that can adversely.”].

Response 3: [….and these parasitic infections can adversely… – … causing parasitic infections that can adversely]. Thank you. In the revised manuscript this change can be found – p. 1; l. 9-10.

Introduction

Comment 4: [Line 25 – Avoid redundancies. Replace “The phylum Apicomplexa includes endoparasites” with “It includes endoparasites.”]

Response 4: [The phylum Apicomplexa includes endoparasites… – It includes endoparasites…]. Thank you. In the revised manuscript this change can be found – p. 1; l. 28.

Comment 5: [Line 56 – Replace “not only in fish but also in different groups of vertebrates” with “not only fish but also different groups of vertebrates.”]

Response 5: […not only in fish but also in different groups of vertebrates… – …not only fish but also different groups of vertebrates…]. Thank you. In the revised manuscript this change can be found – p. 2; l. 58.

 Comment: 6. [Line 61 – Include/correct the numbering of all section headings. Example: 1. Introduction2. Study design3. Adeleorina3.1 Dactylosoma and Babesiosoma, and so on.]

Response: 6. [Thank you. The changes have been done.]

Comment 7: [Line 70, Figure 1 – As already pointed out in the first round of review, in the screening process, the authors must include the reason for excluding papers at each step. The authors state they excluded reports with “Old information,” “Poor quality,” and “Lack of data.” What does this mean? Did the authors apply a time restriction? What was it? What parameters were used to evaluate article quality? What parameters were used to evaluate a lack of data? All this information should be included in detail in the Study Design section. Also, in the figure, replace “SCOPUS” with “Scopus” and “Pub Med” with “PubMed.”]

Response 7: [We added: “The systematic search used in this study initially turned up 747 documents. 432 of these articles were removed based on duplicate records. 111 of the 315 screened records were disqualified due to their abstract, study's similarities. 204 full-length publications were assessed for eligibility, with 53 full-length articles being disqualified for a variety of reasons, old information (28), poor quality (14), lack of data (11). 151 publications were included in the review. Here, we review the known best practices of search strategies, and provide insight on how well recent review and original research articles published in top applied coccidians found in freshwater fish, ensuring the reliability and accuracy of our findings.”] Thank you. In the revised manuscript this change can be found – p. 2; l. 71-79.

Adeleorina

Comment 8: Line 79 – Replace all occurrences of “Schemes” and “Images” with “Figures,” in sequential order as they appear in the document. Examples: Figure 1Figure 2, and so on.

Response 8: [Thank you. The changes have been done.]

Comment: 9. [Line 81 – Replace “the species Apicomplexa” with “the phylum Apicomplexa.”]

Response: 9. [the species Apicomplexa – the phylum Apicomplexa]. Thank you. In the revised manuscript this change can be found – p. 2; l. 86.

Comment 10: Lines 83 to 90 – The text is truncated. Rewrite to make sense.

Response 10: [We have clarified the relationships between the dactylosomatid genera Dactylosoma and Babesiosoma. Species of the genus Babesiosoma are produced in erythrocytes of cold-blooded vertebrates four merozoites, and species of the genus Dactylosoma more than four merozoites during intra-erythrocytic replication [6, 15]. Babesiosoma produces double that number of sporozoites (eight) within oocysts, and Dactylosoma produces up to 16 merozoites and doubles the number of sporozoites within their oocyst in the definitive host [14]. The fish host erythrocyte Babesiosoma differs from Dactylosoma in lacking a nuclear karyosome, possessing vacuolated and only slightly granular cytoplasm, and producing only four merozoites in a rosette or cross [16]. – Distinctive features between two dactylosomatid genera Dactylosoma and Babesiosoma were identified. However, these genera differ according to the number of merozoits (4 merozoits in Babesiosoma, more than 4 in Dactylosoma) that formed in erythrocytes of cold-blooded vertebrates [6, 15]. In definitive host the number of sporozoits within oocysts of both genera is twice as much as a number of merozoits [14]. The fish host erythrocyte Babesiosoma differs from Dactylosoma in lacking a nuclear karyosome, possessing vacuolated and only slightly granular cytoplasm, and producing only four merozoites in a rosette or cross [16].] Thank you. In the revised manuscript this change can be found – p. 2; l. 87-94.

Comment 11: [Line 119 – Replace “concluding that” with “leading to the conclusion that.”]

Response 11: […concluding that… – …leading to the conclusion that…]. In the revised manuscript this change can be found – p. 3; l. 123.

Comment 12: [Lines 120 to 122 – As already pointed out, this paragraph should be moved from here to the end of the section and combined with the paragraph in Lines 150–154 to avoid repetition and redundancies. References should be renumbered, if necessary.]

Response 12: Thank you. [This paragraph was moved]. In the revised manuscript this change can be found – p. 3; l. 153-155.

Comment 13: [Lines 184 to 186 – The sentences are truncated and confusing. Rewrite for clarity.]

Response 13: [There were 23 Haemogregarina species described as freshwater fish. However, some may be placed in Cyrilia, Hepatozoon, or other genera [5]. – It is estimated that some apicomplexans out of 23 Haemogregarina species that described in freshwater fish later placed in Cyrilia, Hepatozoon, or other genera [5].] Thank you. In the revised manuscript this change can be found – p. 4; l. 191-193.

Comment 14: Lines 227 to 234 – The sentences are truncated and confusing. Rewrite for clarity.

Response 14: [The genus Desseria was described in 1995. Members were formerly considered to belong to the genus Haemogregarina. All currently recognized species in this genus infect fish. Although, some early known Desseria species now belong to other genera [7,56,57]. Species of Desseria were described very poorly; only vertebrate stages are known, and often, they are not described in detail. The identity of all these species is doubtful. There is no information about Desseria in the new literature. In 2021, Quraishy et al. recorded the family Haemogregarinidae, which contains the genera Haemogregarina and Cyrilia [58].

– The genus Desseria was described in 1995. All currently recognized species in this genus are infected fish. Species of Desseria were described very poorly; only vertebrate stages are known, and often, they are not described in detail. Some formerly known Deseria species are currently related to other genera [7,56,57]. There is insufficient information about Deseria in contemporary literature. In 2021, the family Haemogregarinidae, which contains the genera Haemogregarina and Cyrilia were suggested by Quraishy et al. [58]]. Thank you. In the revised manuscript this change can be found – p. 4; l. 233-238.

Eimeriorina

Comment 15: Lines 263 to 276 – The sentences are repetitive (with Lines 50 to 56) and/or confusing. Rewrite for conciseness.

Response 15: [Thank you. The changes have been done]. In the revised manuscript this change can be found – p. 5; l. 267-276.

Distribution of coccidian freshwater fish

Comment16: [Lines 414 and 415 – The sentence is confusing, as pointed out before. Rewrite for clarity: “For some species, it was only possible to obtain information regarding the characteristics considered in this review.”]

Response 16: [For some species, obtaining information regarding the characteristics considered in this review was only possible. – For some species, it was only possible to obtain information regarding the characteristics considered in this review] Thank you. In the revised manuscript this change can be found – p. 8; l. 413-414.

Conclusion

Comment 17: [The inclusion of fish pictures in Image 1 was nice. Now, the authors should highlight, in the same image, the main genera of parasites found in each group of fish. This would greatly improve the understanding of the data compiled in the review.]

Response 17: [Thank you. Was done]. In the revised manuscript this change can be found – p. 9; l. 444-457.

Comments on the Quality of English Language

As noted in the first round of revisions, the manuscript has significant language issues that were not fully addressed in this new version.

The authors should consider having the language revised by a native English speaker or using AI tools for English correction. Several truncated sentences and other issues with English writing remain.

Response: [Thank you. We edited the whole MS again.]
